



# A space-time Bayesian hierarchical modeling framework for projection of seasonal streamflow extremes

Álvaro Ossandón[1,2], Manuela I Brunner[3,4], Balaji Rajagopalan[1,5], and William Kleiber[6]

[1]Department of Civil, Environmental and Architectural Engineering, University of Colorado, Boulder CO, USA
[2]Department of Civil Engineering, Santa Maria University, Valparaiso, Chile
[3]Research Applications Laboratory, National Center for Atmospheric Research, Boulder CO, USA
[4]Institute of Earth and Environmental Sciences, University of Freiburg, Freiburg, Germany
[5]Cooperative Institute for Research in Environmental Sciences, University of Colorado, Boulder CO, USA
[6]Department of Applied Mathematics, University of Colorado, Boulder CO, USA

**Correspondence:** Álvaro Ossandón (alvaro.ossandon@colorado.edu)

**Abstract.** Timely projections of seasonal streamflow extremes can be useful for the early implementation of annual flood risk adaptation strategies. However, predicting seasonal extremes is challenging particularly under non-stationary conditions and if extremes are connected in space. The goal of this study is to implement a space-time model for projection of seasonal streamflow extremes that considers the nonstationarity and spatio-temporal dependence of high flows. We develop a space-time model

to project seasonal streamflow extremes for several lead times up to 2 months using a Bayesian Hierarchical Modelling (BHM) framework. This model is based on the assumption that streamflow extremes (3-day maxima) at a set of gauge locations are realizations of a Gaussian elliptical copula and generalized extreme value (GEV) margins with nonstationary parameters. These parameters are modeled as a linear function of suitable covariates from the previous season selected using the deviance information criterion (DIC). Finally, the copula is used to generate streamflow ensembles, which capture spatio-temporal variability

and uncertainty. We apply this modelling framework to predict 3-day maximum flow in spring (May-June) at seven gauges in the Upper Colorado River Basin (UCRB) with 0 to 2 months lead time. In this basin, almost all extremes that cause severe flooding occur in spring as a result of snowmelt and precipitation. Therefore, we use regional mean snow water equivalent and temperature from the preceding winter season as well as indices of large-scale climate teleconnections – ENSO, AMO, and PDO – as potential covariates for 3-day maximum flow. Our model evaluation, which is based on the comparison of different

model versions and the energy skill score, indicates that the model can capture the space-time variability of extreme flow well and that model skill increases with decreasing lead time. We also find that the use of climate variables slightly enhances skill relative to using only snow information. Median projections and their uncertainties are consistent with observations thanks to the representation of spatial dependencies through covariates in the margins and a Gaussian copula. This spatio-temporal modeling framework helps to plan seasonal adaptation and preparedness measures as predictions of extreme spring flows become

available 2 months before actual flood occurrence.





## 1 Introduction

Floods are a concern in mountain regions such as the Upper Colorado River Basin (UCRB), where streamflow extremes happen in spring due to snowmelt in combination with precipitation (McCabe et al., 2007), and are projected to increase under
future climate conditions (Musselman et al., 2018). To reduce the negative impacts of such extreme events, we need tools that decision-makers can use in the mid - and long-term planning of flood risk adaptation strategies. Most existing tools either use hydrological models to provide operational daily forecasts at lead times ranging from 1 day to a couple of weeks or statistical models considering hydro-climatic variables from the previous season to generate seasonal streamflow forecasts. While these tools are useful for reservoir operations during the dry season or high flow alerts at a local scale, they don't usually consider
spatial dependencies in high flow occurrence in different catchments.

Operational streamflow forecasts are generally implemented using physically-based models that use forecasts of hydro-meteorological variables such as rainfall as their forcing (Clark and Hay, 2004; Ghile and Schulze, 2010; Wijayarathne and Coulibaly, 2020). An alternative to such physically based models are hybrid models which combine physically-based models with statistical models to post-process their output and to increase forecast skill (Chen et al., 2015; Kurian et al., 2020). Both
types of models provide daily streamflow forecasts for short lead times (no longer than one or two weeks), may neglect spatial dependencies of flows in different catchments, and are deterministic or provide ensemble forecasts by considering forcing perturbations (of precipitation and temperature), i.e. they don't depict model parameter uncertainty.

Only a few studies have tried to implement seasonal peak flow forecasts, e.g., Werner and Yeager (2013) generated both long- and short-leadtime forecasts during the 2011 runoff season at more than 400 locations in the Colorado River basin using
two physically-based models. However, peak flow forecasts were skillful only after May 15th mainly because of inaccurate weather and climate forecast. In addition, Kwon et al. (2009) generated annual maximum streamflow forecasts for the Three Gorges Dam in the Yangtze River basin in China considering sea surface temperature (SST) anomalies and snow cover from the previous season as covariates. However, these forecasts focused on reservoir operation since they provided return level forecasts for single sites.

Seasonal and sub-seasonal streamflow forecasting models rely on the skill of hydro-climatic variables from the previous season, such as snow cover (e.g., Kwon et al., 2009; Livneh and Badger, 2020; Pagano et al., 2009), large scale climate indices (Lima and Lall, 2010; Robertson and Wang, 2012; Ruiz et al., 2007), or changes in land cover conditions (Penn et al., 2020) among others to obtain skillful forecasts. Modelling approaches span statistical approaches based on multiple linear regression (Pagano et al., 2009; Penn et al., 2020; Ruiz et al., 2007); physically-based models that consider the uncertainty of initial
conditions or inputs by perturbing them (Anghileri et al., 2016; Wood et al., 2016; Werner and Yeager, 2013); and Bayesian approaches that account for parameter uncertainty (Kwon et al., 2009; Lima and Lall, 2010; Robertson and Wang, 2012).

In the case of mountain regions, projection models for spring flow extremes can take advantage of the fact that snow water equivalents (SWE) accumulated until the beginning of spring are a skillful predictor of spring streamflow (Koster et al., 2010; Livneh and Badger, 2020). However, such models should also consider potential nonstationarities due to climate change
(Musselman et al., 2018) and the spatio-temporal dependence of the data as flood risk may increase if multiple catchments





are affected at once (Hochrainer-Stigler, 2020). We here develop a Bayesian Hierarchical Model (BHM) for the prediction of extreme spring flow that considers both non-stationarity and spatio-temporal dependence and ask:

– How does the representation of nonstationarity through suitable covariates improve seasonal predictions?

– How does the explicit representation of spatial dependencies improve prediction performance?

– To what extent are seasonal projections for longer lead times still skillful?

We address these question by applying the proposed BHM to project 3-day maximum spring (May-June) flow at seven gauges in the Upper Colorado River Basin (UCRB).

## 2  Proposed framework

We propose a space-time modeling framework for the prediction of seasonal streamflow extremes that has three components-
(i) a hierarchical model structure, (ii) nonstationary margins, and (iii) a spatial dependence model. Each of these model components and their estimation strategies and the estimation of ensembles of seasonal streamflow extremes are described below.

### 2.1  Hierarchical model structure

We conduct a nonstationary frequency analysis of seasonal streamflow extremes at $m$ gauges in a river basin - say, $q_1, \ldots, q_m$ - over $k$ years. To do this, we consider a Bayesian hierarchical model that accounts for spatial dependence and nonstationarity.
In the first layer of the hierarchical model structure, also known as the data layer, we consider that the joint distribution of streamflow at multiple gauges in each year is modeled using a Gaussian elliptical copula with generalized extreme value (GEV) margins (Coles, 2001; He et al., 2015; Katz, 2013). Specifically, the proposed model structure for $m$ locations is

$$(q_1(t), \ldots, q_m(t)) \sim C_m \left(\mathbf{\Sigma}; \boldsymbol{\mu}(t), \boldsymbol{\sigma}(t), \boldsymbol{\xi}\right) \tag{1}$$

$$q_i(t) \sim GEV\left(\mu_i(t), \sigma_i(t), \xi_i\right), \quad i = 1, \ldots, m \tag{2}$$

where $C_m$ is an m-dimensional Gaussian elliptical copula with dependence matrix $\mathbf{\Sigma}$ and GEV parameters (location, scale, and shape) $\mu_i(t) \in (-\infty, \infty)$, $\sigma_i(t) > 0$ and $\xi_i \in (-\infty, \infty)$. The second layer of the hierarchy, also known as the latent layer, considers that the three parameters are nonstationary and are time-varying, i.e., for the streamflow gauge $i$ at time $t$, $\boldsymbol{\mu}(t) = [\mu_i(t)]_{i=1}^m$, $\boldsymbol{\sigma}(t) = [\sigma_i(t)]_{i=1}^m$, $\boldsymbol{\xi} = [\xi_i]_{i=1}^m$. Specifics of models (1) and (2) are given in the next sections.

A conceptual sketch of the BHM is shown in Fig. 1 which shows the data layer (Gaussian copula and GEV marginal
distributions) and the latent layer (time dependence of GEV parameters).



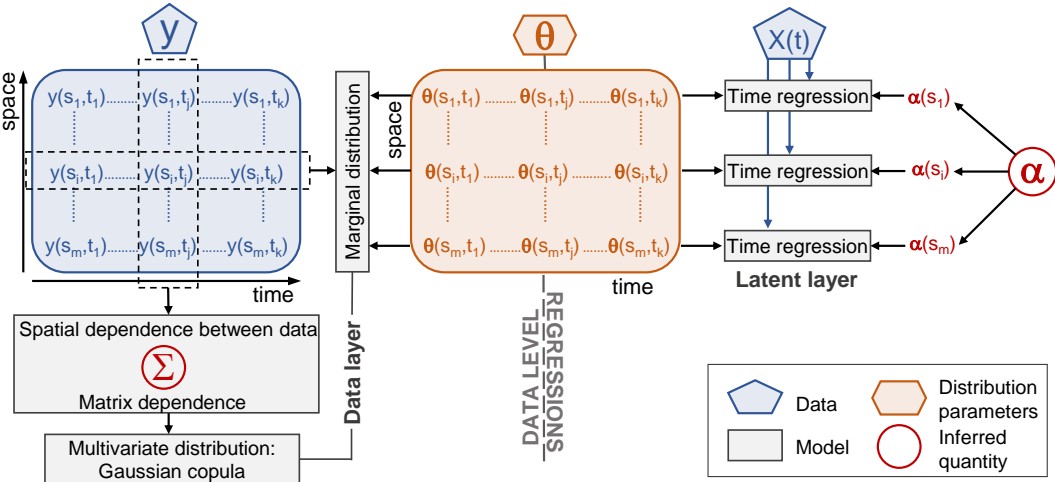

**Figure 1.** Conceptual sketch of the Bayesian Hierarchical Model. Blue boxes denote the data, orange boxes the GEV distribution parameters, grey boxes the models, and red circles the inferred quantities. Model boxes correspond to the data layer (Gaussian copula and GEV marginal distributions) and the latent layer (time regressions of GEV parameters). $\boldsymbol{\theta}(s_i, t_j) = [\boldsymbol{\mu}(s_i, t_j), \log \boldsymbol{\sigma}(s_i, t_j), \boldsymbol{\xi}(s_i, t_j)]$.

## 2.2 Marginal distributions

The first two GEV parameters (location and scale) are modeled as linear functions of time-dependent large-scale climate variables and regional mean variables from the previous season and the shape parameter is considered to be stationary:

$$\mu_i(t) = \alpha_{\mu 0_i} + \sum_{j=1}^{n} \alpha_{\mu j_i} x_j(t), \quad i = 1, \ldots, m \tag{3}$$

$$\log(\sigma_i(t)) = \alpha_{\sigma 0_i} + \sum_{j=1}^{n} \alpha_{\sigma j_i} x_j(t), \quad i = 1, \ldots, m \tag{4}$$

$$\xi_i = \alpha_{\xi 0_i}, \quad i = 1, \ldots, m \tag{5}$$

where $x_j(t)$ is covariate $j$ at time $t$, $\alpha_{\mu j_i}$, $\alpha_{\sigma j_i}$, and $\alpha_{\xi 0_i}$ are the regression coefficients for covariate $j$ and gauge $i$ at time $t$. $\log(\sigma)$ is used to ensure positive scale parameters. The validity of the nonstationary location and scale parameters can be checked through the significance of their slope coefficients' posterior PDFs ($\boldsymbol{\alpha}_{\sigma_j} \neq 0$, and $\boldsymbol{\alpha}_{\xi_j} \neq 0$ for $j > 0$). Covariates
will be discussed in section 3.2. The shape parameter $\xi$ is often modeled as a single value per study area/region because of large estimation uncertainties (Apputhurai and Stephenson, 2013; Atyeo and Walshaw, 2012; Cooley et al., 2007; Renard, 2011) or modelled spatially along with the other GEV parameters but considering a specific range of potential values (Bracken et al., 2016; Cooley and Sain, 2010). Here, we modeled the shape parameter for each catchment individually but considered it stationary in time.





## 2.3 Gaussian copula for spatial dependence


Copulas are a flexible tool for modeling multivariate random variables since they can represent dependence independently of the choice of marginal distributions (Bracken et al., 2018; Brunner et al., 2019; Genest and Favre, 2007), which is a particularly appealing feature for extremal processes. This study focuses on Gaussian copulas because of their ease of implementation in a Bayesian and high-dimensional framework.

Let $\boldsymbol{q}(t) = [q_i(t)]_{i=1}^m$ be a random vector of extreme streamflow at $m$ gauges and time $t$. The Gaussian copula builds the joint cumulative distribution function (CDF) of $\boldsymbol{q}(t)$ as

$$F_{cop}(\boldsymbol{q}(t)) = \Phi_\Sigma(\boldsymbol{u}(t)) \tag{6}$$

where $\Phi_\Sigma(\cdot)$ is the joint CDF of an $m$-dimensional multivariate normal distribution with dependence matrix $\Sigma$, $u(t) = [u_i(t)]_{i=1}^m$, $u_i(t) = \Phi^{-1}(F_{it}[q_i(t)])$ with $\Phi$ being the CDF of the standard normal distribution, and $F_{it}(\cdot)$ the marginal GEV
or empirical CDF for streamflow gauge $i$ at time $t$.

The copula dependence matrix, $\Sigma$, is a symmetric positive definite matrix that captures the strength of dependence between all gauge pairs using the Pearson correlation coefficient. The element $c_{ij}$ of $\Sigma$ quantifies the dependence between gauges $i$ and $j$, and its values can vary between -1 and 1:

$$\Sigma = \begin{bmatrix} 1 & c_{12} & \cdots & c_{1(m-1)} & c_{1m} \\ c_{21} & 1 & \ddots & \vdots & c_{2m} \\ c_{31} & c_{32} & \ddots & \vdots & \vdots \\ \vdots & \vdots & \ddots & 1 & c_{(m-1)m} \\ c_{m1} & c_{m2} & \cdots & c_{(m-1)m} & 1 \end{bmatrix} \tag{7}$$

By definition, the dependency between a streamflow gauge and itself is unity, so the diagonal elements of $\Sigma$ are 1's. Since $\Sigma$ is symmetric, only $m(m-1)/2$ values need to be fitted (values in the lower or upper triangle of $\Sigma$). The Gaussian copula only assumes linear correlation after quantile transformation of the marginals with the inverse normal CDF. This does not impose a linear correlation structure on the marginal distributions, meaning that nonlinear dependence between variables can be captured at the data level (Bracken et al., 2018).

There are two main approaches to estimate the unknown parameters of the conditional copula (Hochrainer-Stigler, 2020). The first one is called inference functions for margins (IFM). In this approach, the marginal distribution parameters are estimated in the first step and the copula parameters in the second step. One major disadvantage of this approach is the loss of estimation efficiency, as in the first step, the dependence between the marginal distributions is not considered. The second approach (hereafter referred to as pseudo-observations fitting) estimates the copula parameters without assuming specific
parametric distribution functions of the marginals, and pseudo-observations are used instead. The framework proposed here considers pseudo-observations fitting to estimate the copula dependence parameters.



## 2.4 Estimation strategy

Inference of the model parameters is done in a Bayesian framework, which can account for parameter uncertainties. The posterior distributions of the model parameters, $\boldsymbol{\theta} = [\boldsymbol{\mu}, \log\boldsymbol{\sigma}, \boldsymbol{\xi}]$ and $\boldsymbol{\Sigma}$, given the data (3-day maximum spring (May-June) flow at each gauge and covariates) and considering a record length of $k$ days by Bayes' rule, are

$$p(\boldsymbol{\theta}, \boldsymbol{\Sigma}|\boldsymbol{q}, \boldsymbol{x}, \boldsymbol{u}) \propto \left[\prod_{t=1}^{k}\left(\prod_{i=1}^{m} p_q\left(q_i(t)|\theta_i, \boldsymbol{x}(t)\right)\right) p_u\left(\boldsymbol{u}(t)|\boldsymbol{\Sigma}\right)\right] p_\theta\left(\boldsymbol{\theta}\right) p_\Sigma\left(\boldsymbol{\Sigma}\right) \tag{8}$$

where $p_q\left(q_i(t)|\theta_i, \boldsymbol{x}(t)\right)$ represents the PDF of the GEV for location $i$ and time $t$, $p_u\left(\boldsymbol{u}(t)|\boldsymbol{\Sigma}\right) = MVN\left(\boldsymbol{u}(t)|\boldsymbol{0}, \boldsymbol{\Sigma}\right)$ is the Gaussian copula PDF for time $t$, $u(t) = [u_i(t)]_{i=1}^{m}$, $u_i(t) = \Phi^{-1}\left(F_{it}[q_i(t)]\right)$, $\Phi$ is the CDF of the standard normal distribution, $F_{it}(\cdot)$ is the empirical CDF (pseudo-observations) for location $i$ at time $t$, and $p_\Sigma\left(\boldsymbol{\Sigma}\right)$ and $p_\theta\left(\boldsymbol{\theta}\right)$ represent the priors of the GEV regression coefficients and Gaussian copula dependence matrix, respectively. $p_\theta\left(\boldsymbol{\theta}\right)$ is defined as

$$p_\theta\left(\boldsymbol{\theta}\right) = p_\mu\left(\boldsymbol{\mu}\right) p_\sigma\left(\log\boldsymbol{\sigma}\right) p_\xi\left(\boldsymbol{\xi}\right) \tag{9}$$

$$p_\mu\left(\boldsymbol{\mu}\right) = \prod_{j=0}^{n} MVN\left(\boldsymbol{\alpha}_{\mu_j}\middle|\boldsymbol{0}, \boldsymbol{\Sigma}_{\alpha_{\mu_j}}\right) p_{\Sigma_{\alpha_{\mu_j}}}\left(\boldsymbol{\Sigma}_{\alpha_{\mu_j}}\right) \tag{10}$$

$$p_\sigma\left(\log\boldsymbol{\sigma}\right) = \prod_{j=0}^{n} MVN\left(\boldsymbol{\alpha}_{\sigma_j}\middle|\boldsymbol{0}, \boldsymbol{\Sigma}_{\alpha_{\sigma_j}}\right) p_{\Sigma_{\alpha_{\sigma_j}}}\left(\boldsymbol{\Sigma}_{\alpha_{\sigma_j}}\right) \tag{11}$$

$$p_\xi\left(\boldsymbol{\xi}\right) = MVN\left(\boldsymbol{\alpha}_{\xi_0}\middle|\boldsymbol{0}, \boldsymbol{\Sigma}_{\alpha_{\xi_0}}\right) p_{\Sigma_{\alpha_{\xi_0}}}\left(\boldsymbol{\Sigma}_{\alpha_{\xi_0}}\right), \tag{12}$$

where $MVN\left(\boldsymbol{\alpha}_{\mu_j}\middle|\boldsymbol{0}, \boldsymbol{\Sigma}_{\alpha_{\mu_j}}\right)$, $MVN\left(\boldsymbol{\alpha}_{\sigma_j}\middle|\boldsymbol{0}, \boldsymbol{\Sigma}_{\alpha_{\sigma_j}}\right)$, and $MVN\left(\boldsymbol{\alpha}_{\xi_0}\middle|\boldsymbol{0}, \boldsymbol{\Sigma}_{\alpha_{\xi_0}}\right)$ represent probability densities of multivariate normal distributions with mean $\boldsymbol{0}$ and covariance matrices $\boldsymbol{\Sigma}_{\alpha_{\mu_j}}$, $\boldsymbol{\Sigma}_{\alpha_{\sigma_j}}$, and $\boldsymbol{\Sigma}_{\alpha_{\xi_0}}$ correspond to the priors of the GEV regression coefficients $\boldsymbol{\alpha}_{\mu_j}$, $\boldsymbol{\alpha}_{\sigma_j}$, and $\boldsymbol{\alpha}_{\xi_0}$ respectively; and $p_{\Sigma_{\alpha_{\mu_j}}}\left(\boldsymbol{\Sigma}_{\alpha_{\mu_j}}\right)$, $p_{\Sigma_{\alpha_{\sigma_j}}}\left(\boldsymbol{\Sigma}_{\alpha_{\sigma_j}}\right)$, and $p_{\Sigma_{\alpha_{\xi_0}}}\left(\boldsymbol{\Sigma}_{\alpha_{\xi_0}}\right)$ are the priors of $\boldsymbol{\Sigma}_{\alpha_{\mu_j}}$, $\boldsymbol{\Sigma}_{\alpha_{\sigma_j}}$, and $\boldsymbol{\Sigma}_{\alpha_{\xi_0}}$, which based on Gelman and Hill (2006) are assumed to follow an inverse-Wishart distribution to ensure a positive definite covariance matrix

$$\boldsymbol{\Sigma}_{\alpha_{\mu_j}} \sim Inv\ wishart\left(\nu, A_j \boldsymbol{I}\right) \tag{13}$$

$$\boldsymbol{\Sigma}_{\alpha_{\sigma_j}} \sim Inv\ wishart\left(\nu, B_j \boldsymbol{I}\right) \tag{14}$$

$$\boldsymbol{\Sigma}_{\alpha_{\xi_0}} \sim Inv\ wishart\left(\nu, C_0 \boldsymbol{I}\right). \tag{15}$$

$\nu$ corresponds to the degrees of freedom $(m+1)$, $\boldsymbol{I}$ is an $(m+2)\times(m+2)$ identity matrix, and $A_j$, $B_J$, and $C_0$ are scalars properly set for $\boldsymbol{\Sigma}_{\alpha_{\mu_j}}$, $\boldsymbol{\Sigma}_{\alpha_{\sigma_j}}$, and $\boldsymbol{\Sigma}_{\alpha_{\xi_0}}$, respectively. The regression coefficients are modeled jointly to capture their spatial correlations. Finally, $p_\Sigma\left(\boldsymbol{\Sigma}\right)$ represents the prior of the Gaussian copula dependence matrix and is also assumed to follow an inverse-Wishart distribution to ensure a positive definite covariance matrix

$$\boldsymbol{\Sigma} \sim Inv\ wishart\left(\nu, COV_{\boldsymbol{u}}\right) \tag{16}$$





where $COV_u$ corresponds to the covariance matrix of the uniform quantiles obtained from the pseudo-observations.

### 2.5 Estimation of ensembles of seasonal streamflow projections

The predictive posterior distribution of spring maximum streamflow (ensembles) for the $m$ streamflow gauges can be estimated using the posterior distributions ($M$ samples) of GEV regression coefficients, $\boldsymbol{\alpha}_{\mu_j}$, $\boldsymbol{\alpha}_{\sigma_j}$, and $\boldsymbol{\alpha}_{\xi_0}$, and the Gaussian copula dependence matrix, $\boldsymbol{\Sigma}$, have been estimated using the estimation strategy presented in the previous section. The steps for this procedure are as follows:

1. Select a single posterior sample of all model parameters ($\boldsymbol{\alpha}_{\mu_j}$, $\boldsymbol{\alpha}_{\sigma_j}$, $\boldsymbol{\alpha}_{\xi_0}$, and $\boldsymbol{\Sigma}$).

2. If posterior PDFs of slope coefficients of GEV parameters were found to be significant, compute the GEV parameters ($\mu_i(t)$, $\log(\sigma_i(t))$) for each year and gauge using $\boldsymbol{\alpha}_{\mu_i}$, $\boldsymbol{\alpha}_{\sigma_i}$, and covariates, $\boldsymbol{x}(t)$, using Eq. (3)-(4).

3. Simulate marginal cumulative distributions, $F_{it}$, for the $m$ streamflow gauges from a Gaussian copula with a dependence matrix, $\boldsymbol{\Sigma}$.

4. Compute spring maximum streamflow for each streamflow gauge $i$ at time $t$ using the following expression

$$q_i(t) = \mu_i(t) + \frac{\sigma_i(t)}{\xi_i}\left[(-\log(F_{it}))^{-\xi_i} - 1\right], \quad i = 1, \ldots, m \tag{17}$$

where $\sigma_i(t) = \exp(\log(\sigma_i(t))$

5. Repeat steps 2-4 for each gauge and year of the record.

6. Repeat steps 1-5 for each posterior sample, i.e., $M$ times.

## 3 Application to the Upper Colorado River basin

We demonstrate the utility of the framework proposed in the previous section by applying it to project 3-day maximum spring (May-June) flow at seven gauges in the Upper Colorado River basin (UCRB) (Fig. 2). The UCRB is located in west-central Colorado, US, and has an area of approximately 25,700 km². Its headwaters originate at the Continental Divide in Rocky Mountain National Park. Further downstream, UCRB flows in a westerly direction through forested mountains and irrigated valleys before it leaves Colorado in Mesa County downstream of the City of Grand Junction (Colorado's Decision Support

Systems, 2007). The basin has a snowmelt-dominated flow regime, i.e., most of the precipitation is accumulated during winter in the form of snow. Over 85% of the basin's streamflow and flood peaks occur in spring due to snowmelt. Long lead projections of spring flow extremes allow water resources managers to develop robust flood plain preservation and flood mitigation strategies.

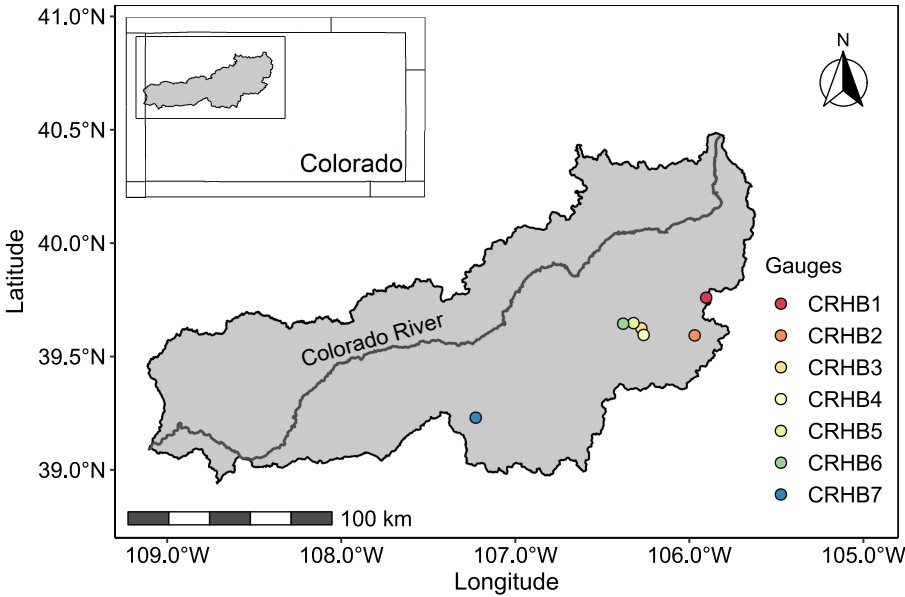

**Figure 2.** Streamflow gauges in the Upper Colorado River basin (UCRB) considered in this study.

### 3.1 Streamflow data

Daily spring, May through June, streamflow data were obtained from the US Geological Survey (USGS) using the R package dataRetrieval (De Cicco et al., 2018). We selected streamflow gauges located inside the UCRB with no more than 10% or three consecutive years of missing data from 1965 to 2018. This procedure resulted in the selection of seven streamflow gauges (Fig. 2 and Table 1). The seven gauges have drainage areas between 14.5 and 432.5 $km^2$, elevations between 2105 and 3179 m, and mean streamflow and mean seasonal (May-June) streamflow range from 1 to 28.3 $m^3\,s^{-1}$ and from 0.3 to 8.3 $m^3\,s^{-1}$,

respectively. We computed 3-day maximum spring streamflow for each year at each streamflow gauge. For a gauge with missing annual values, these values were substituted with the gauge's median value.

In addition, Figure 3 shows the spatial dependence of 3-day maximum spring streamflow for the seven gauges, which was assessed using Kendall's rank correlation for each pair of stations. There is generally a positive dependence between 3-day maximum spring flows between gauges, as indicated by Kendall's rank correlation values higher than 0.4 (significant at 95%

confidence level) for all station pairs. These significant spatial correlations require the inclusion of a copula in the BHM to capture the spatial dependence of the data.

### 3.2 Covariates

It has been shown in previous studies that accumulated snow water equivalent (SWE) until the beginning of spring is the most skillful predictor of spring-summer seasonal streamflow across mountainous regions, such as the western US (Koster

et al., 2010; Livneh and Badger, 2020; Pagano, 2010; Wood et al., 2016). For the Colorado River, strong links between annual





**Table 1.** Basic data corresponding to the streamflow gauges in the Upper Colorado River basin (UCRB) considered in this study.

| Code | USGS gauge number | River | Drainage area ($km^2$) | Elevation (m) | Mean streamflow ($m^3 s^{-1}$) | Mean seasonal streamflow ($m^3 s^{-1}$) |
|---|---|---|---|---|---|---|
| UCRB1 | 9034900 | Bobtail creek | 14.5 | 3179 | 1.0 | 0.3 |
| UCRB2 | 9047700 | Keystone gulch | 23.7 | 2850 | 0.5 | 0.2 |
| UCRB3 | 9065500 | Gore creek | 37.6 | 2621 | 3.1 | 0.8 |
| UCRB4 | 9066000 | Black gore creek | 32.4 | 2789 | 2.0 | 0.5 |
| UCRB5 | 9066200 | Booth creek | 16.0 | 2537 | 1.3 | 0.3 |
| UCRB6 | 9066300 | Middle creek | 15.4 | 2499 | 0.7 | 0.2 |
| UCRB7 | 9081600 | Crystal river | 432.5 | 2105 | 28.3 | 8.3 |

| | | | | | | |
|---|---|---|---|---|---|---|
| UCRB1 | 0.53 | 0.51 | 0.55 | 0.56 | 0.63 | 0.53 |
| | UCRB2 | 0.4 | 0.68 | 0.43 | 0.55 | 0.42 |
| | | UCRB3 | 0.54 | 0.61 | 0.53 | 0.48 |
| | | | UCRB4 | 0.51 | 0.64 | 0.57 |
| | | | | UCRB5 | 0.61 | 0.43 |
| | | | | | UCRB6 | 0.53 |
| | | | | | | UCRB7 |

**Figure 3.** Pairs plot (lower triangular matrix) and Kendall's rank correlation coefficients (upper triangular matrix) of spring 3-day maximum streamflow. Kendall's rank correlation is significant (P-value < 0.1), and positive association patterns are visible for all pairs.

flow and large scale climate drivers have been documented – e.g., with AMO (Enfield et al., 2001; Hidalgo, 2004; McCabe and Dettinger, 1999; McCabe et al., 2007; Nowak et al., 2012; Timilsena et al., 2009; Tootle et al., 2005), PDO (Hidalgo, 2004; Timilsena et al., 2009; Tootle et al., 2005), and ENSO (Kahya and Dracup, 1994; Rajagopalan et al., 2000; Redmond and Koch, 1991; Thomson et al., 2003; Timilsena et al., 2009). However, the relationship of these indices with annual or





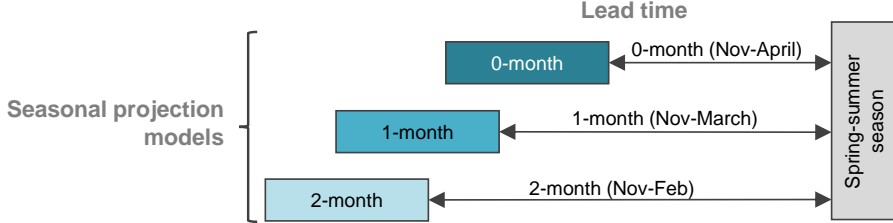

**Figure 4.** Schemes of nonstationary models considered for three different lead times. The dark turquoise box denotes the model for a 0-month lead time (projections are released on May 1st), the turquoise box the model for a 1-month lead time (projections are released on April 1st), and the light turquoise box the model for a 2-month lead time (projections are released on March 1st). The same color scheme will be considered for the results section.

seasonal extremes have not been explored in-depth and earlier studies showed modest relationships – e.g. between Southwest US seasonal or annual maximum streamflow and ENSO and PDO (Werner et al., 2004; Sankarasubramanian and Lall, 2003).

We will still test the usefulness of climatic drivers within the BHM framework by using them as potential covariates of the marginal GEV distributions. Specifically, we tested the following covariates for modeling the temporal nonstationarity of the GEV parameters (see Eq. (3)-(4)) for the period 1965-2018: average El Nino Southern Oscillation (ENSO), Northern Pacific

– Pacific Decadal Oscillation (PDO), and Atlantic – Atlantic Multidecadal Oscillation (AMO) climate indices, spatial average of accumulated snow water equivalent (SASWE) from November to February, March, or April depending on the lead time (0, 1 or 2 months lead time), and spatial average April mean temperature (SAAMT). As indicated earlier, we focus on projecting 3-day extreme spring flow at lead times of 0-, 1- and 2- months lead time corresponding to forecast issuance on May 1st, April 1st and March 1st, respectively. This is shown in Fig. 4. The covariates use all information prior to forecast issuance.

The potential covariates – monthly ENSO, PDO, and AMO climate indices, monthly accumulated snow water equivalent, SWE, from November to April, and daily mean temperature for April – were computed using the following data sources: We obtained ENSO, PDO, and AMO climate indices values from the National Oceanic and Atmospheric Administration (NOAA; https://psl.noaa.gov/data/climateindices/list/). We obtained monthly accumulated snow water equivalent (SWE) values from the Natural Resources Conservation Service (NRCS) Snow Course (https://wcc.sc.egov.usda.gov/reportGenerator/). April's daily

mean temperature was downloaded from the Global Historical Climatology Network (GHCN; Menne et al., 2012). For SWE and daily mean temperature, we only considered stations located inside the UCRB. We then computed SASWE and SAAMT covariates as the simple mean of UCRB station values for each year. We considered the same SWE covariate (SASWE) for all the gauges because most of the snow courses in the UCRB are located above the locations of the first six station gauges, which are located close to each other (see Fig. 2).

We assessed the strength of the relationship between the covariates and maximum spring flow by computing the Spearman's rank correlation coefficient, which is shown in Fig. 5 for a 0-month lead time (covariates were calculated from November to April). It can be seen that SASWE (Fig. 5a) exhibits a significant and strong positive correlation with spring maximum streamflow at all the gauges, which was expected and also supports the choice of SASWE as the covariate for all the station





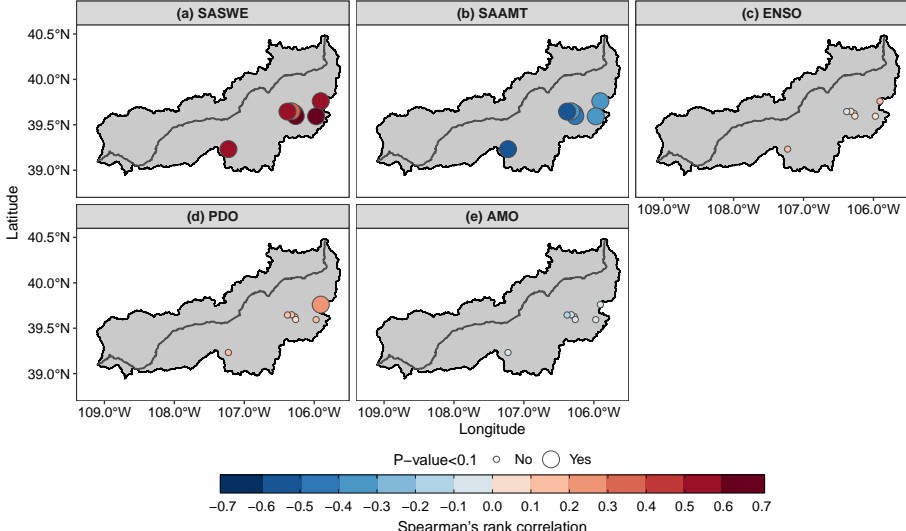

**Figure 5.** Spearman's rank correlation coefficient between 3-day maximum spring flow and potential covariates for a 0-month lead time: (a) the spatial average of Nov-April snow water equivalent (SASWE) over the UCRB; (b) the spatial average of April mean temperature (SAAMT) over the UCRB; (c) mean Nov-April ENSO; (d) mean Nov-April PDO; and (e) mean Nov-April AMO. Big circles indicate that Spearman's rank correlation is significant (p-value < 0.1).

gauges. At the same time, SAAMT (Fig. 5b) shows a significant negative correlation with maximum spring flow. This finding

was also expected since a higher mean April temperature would lead to early melt and therefore less snow availability for melt later in spring. The ENSO, PDO, and AMO climate indices (Fig. 5c-e ) show a weaker correlation with maximum spring flow than SASWE and SAAMT at almost all the gauges. Similar correlations were found for other lead times (Figs. A1 and A2). For each lead time, we obtained the best nonstationary GEV model as the combination of predictors that resulted in a minimum deviance information criterion (DIC; Spiegelhalter et al., 2002). The DIC corresponds to a hierarchical modeling

generalization of the Akaike information criterion (AIC; Akaike, 2011) and facilitates Bayesian model selection. The DIC is computed for a suite of candidate models with various combinations of covariates and the model with the minimum DIC is selected for predicting maximum spring flow in the UCRB.

### 3.3 Adequacy of the GEV distribution and Gaussian copula

We checked the adequacy of the GEV distribution as a marginal and the Gaussian copula to capture spatial dependence using

their maximum likelihood estimates for 3-day maximum spring flow.

To check the validity of the GEV distribution as a marginal distribution, we fitted a stationary GEV distribution at each gauge using maximum likelihood. Then, we ran two goodness-of-fit tests, the Cramer-von Mises and Anderson-Darling tests (D'Agostino and Stephens, 1986). The p-values for the two tests were higher than 0.3 for most of the gauges, except for UCRB1 (0.11), which implies that at the 95% confidence level, there is insufficient evidence to reject the null hypothesis that the data





come from a GEV distribution. Q-Q plots of the stationary GEV distribution fitted for spring 3-day maximum streamflow for

seven streamflow gauges of UCRB can be found in Fig. A3.

For testing the adequacy of the Gaussian copula, we ran three multivariate normality tests using marginal transformations based on pseudo-observations: Royston's (Royston, 1982), Mardia's (Mardia, 1970), and Henze-Zirkler's tests (Henze and Zirkler, 1990). The p-values obtained for the three tests were 0.27, 0.45, and 0.6, respectively, which did not reject the null

hypothesis that the transformed data follow a multivariate normal distribution. As an illustration of the goodness fit of the Gaussian copula, Figure 6 shows the pairwise dependence structure between gauges UCRB2 and UCRB4 for different copula families (for more details about the other types of copulas, the reader is referred to Hochrainer-Stigler (2020) or Genest and Favre (2007)). Black points in each panel (Fig. 6b-g) correspond to the observed CDF values. The simulated contour lines represent simulated contour lines for different dependence structures, including independence (no copula, Fig. 6b), Gaussian

(Fig. 6c), Student-t (Fig. 6d), Joe (Fig. 6e), Gumbel (Fig. 6f), and Vine copulas (Fig. 6g). Only Gaussian, Student-t, and Vine copulas can capture the dependence structure of the data. This visual inspection confirms that the Gaussian copula is suitable to replicate the dependence structure of the observed data.

### 3.4 Model structure for the UCRB

The specific structure of the BHM for the UCRB incorporated the covariates described in section 3.2. We modeled the location

parameter of the GEV at each gauge in a nonstationary way but the scale parameter of the GEV was kept stationary for all gauges. An initial run of the BHM with a nonstationary scale parameter showed that the regression coefficients ($\boldsymbol{\alpha}_{\sigma_j} = 0$ for $j > 0$ in Eq.(4)) were not significant (posterior PDFs contain zero in the 95% credible interval). The priors of the covariance matrix of $\boldsymbol{\alpha}_{\mu_j}$, $\boldsymbol{\alpha}_{\sigma_0}$, and $\boldsymbol{\alpha}_{\xi_0}$ used are:

$$\boldsymbol{\Sigma}_{\boldsymbol{\alpha}_{\mu_0}} \sim Inv\,wishart\,(8, A_0 \boldsymbol{I}) \tag{18}$$

$$\boldsymbol{\Sigma}_{\boldsymbol{\alpha}_{\mu_j}} \sim Inv\,wishart\,(8, A_j \boldsymbol{I}), \quad j \geq 1 \tag{19}$$

$$\boldsymbol{\Sigma}_{\boldsymbol{\alpha}_{\sigma_0}} \sim Inv\,wishart\,(8, B_0 \boldsymbol{I}) \tag{20}$$

$$\boldsymbol{\Sigma}_{\boldsymbol{\alpha}_{\xi_0}} \sim Inv\,wishart\,(8, C_0 \boldsymbol{I}) \tag{21}$$

We considered noninformative priors for the covariance matrix of the GEV regression coefficients by setting $A_0 = 100$, $A_j = 10$, $B_0 = 1$, and $C_0 = 1$. Since we fitted different types of BHMs, $\boldsymbol{\Sigma}_{\boldsymbol{\alpha}_j}$ is only considered for nonstationary models. For

the dependence matrix of the Gaussian copula, we used an informative prior

$$\boldsymbol{\Sigma} \sim Inv\,wishart\,(8, COV_{\boldsymbol{u}}) \tag{22}$$

where $COV_{\boldsymbol{u}}$ corresponds to the covariance matrix of the uniform quantiles obtained from the pseudo-observations of maximum spring flow.



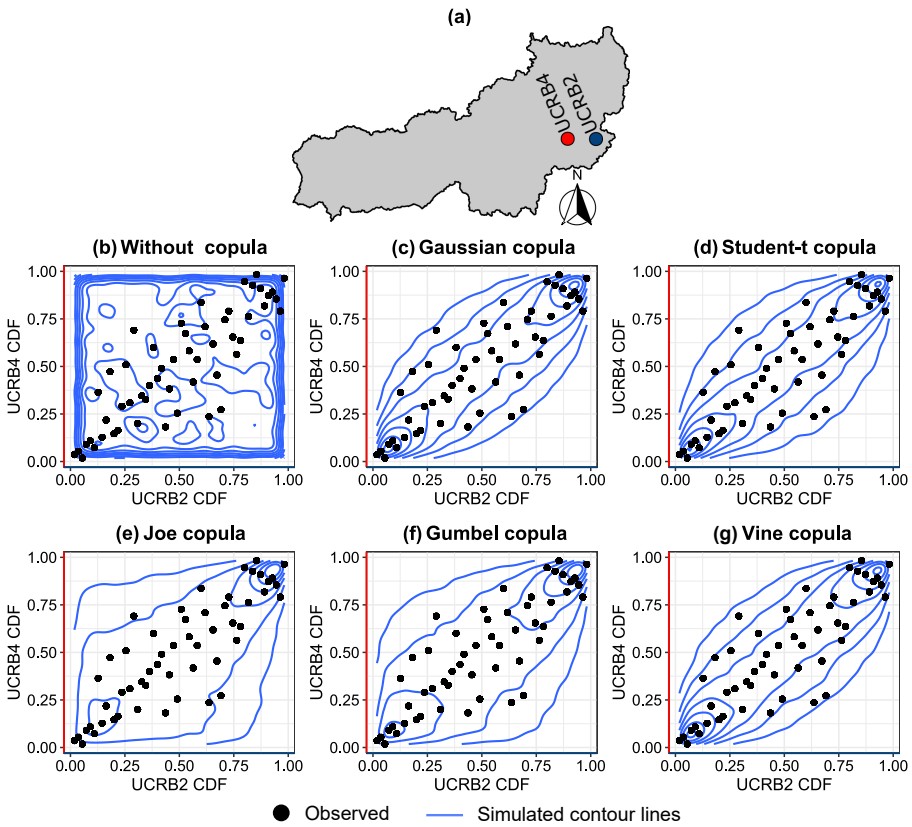

**Figure 6.** Pairwise dependence structure between UCRB2 and UCRB4 (a) streamflow simulated (b) without a copula and with a (c) Gaussian copula; (d) Student-t copula; (e) Joe copula; (f) Gumbel copula; and (g) Vine copula. Observations and simulated contour lines are shown as black points and blue lines, respectively.

## 3.5 Implementation and model fitting

The model was implemented in R (R Core, 2017) using the program JAGS (Just Another Gibbs Sampler; Plummer, 2003) and the R package rjags (Plummer, 2019), which provides an interface from R to the JAGS library for Bayesian data analysis. Posterior distributions of the GEV regression coefficients and Gaussian copula matrix were estimated using Gibbs sampling (Gelman and Hill, 2006; Robert and Casella, 2011) based on the priors assigned in the previous section. The predictive posterior distributions of maximum spring flow (ensembles) for all years were estimated according to section 2.5. We ran three parallel

chains with different initial values and each simulation was performed for 2,000,000 iterations with a burn-in size value of 1,000,000 to ensure convergence. To reduce the sample dependence (autocorrelation), the thinning factor was set to 500, which yields 6000 samples (2000 samples from each chain). The scale reduction factor $\hat{R}$ (Gelman and Rubin, 1992) was used to check for model convergence, i.e., $\hat{R}$ values less than the critical value of 1.1 suggest adequate convergence of the model. In all of our runs, the $\hat{R}$ values were less than 1.1 in 6000 samples, indicating convergence. Consequently, the posterior distributions





of the GEV regression coefficients, the Gaussian copula matrix, and the predictive posterior distributions of maximum spring
flow consisted of 6000 ensembles.

## 3.6 Model cross-validation and verification metrics

To test the out-of-sample predictability of the model, we performed leave-one-year-out cross-validation by dropping one year
from the record (1965–2018) and fitting the BHM using the remaining years, also known as the calibration years. The fitted
model is applied to provide estimates for the one validation year. This cross-validation procedure was repeated 54 times.

As the goal of this study is to provide seasonal streamflow extremes projections for risk-based flood adaptation, we also
implemented this leave-one-year-out cross-validation just for high-flow years in which all the station gauges exceeded their
60th percentile.

We computed the energy skill score (ESS) as our verification metric since we are interested in capturing the spatio-temporal
dependence of the data. The energy score (ES) assesses probabilistic forecasts of a multivariate quantity (Gneiting et al., 2008;
Gneiting and Raftery, 2007):

$$\text{ES} = \frac{1}{M} \sum_{j=1}^{M} ||\boldsymbol{q}_j - \boldsymbol{q}_o|| - \frac{1}{2M^2} \sum_{i=1}^{M} \sum_{j=1}^{M} ||\boldsymbol{q}_i - \boldsymbol{q}_j|| \tag{23}$$

where $M = 3000$ is the size of the ensemble forecast, $\boldsymbol{q}_j$ is the $7 \times 1$ vector of the $j$th ensemble forecast at year $t$, $\boldsymbol{q}_o$
is the $7 \times 1$ vector of observed streamflow at year $t$, and $\cdot$ denotes the Euclidean norm. This performance metric is a direct
generalization of the continuous ranked probability score (CRPS; Gneiting et al., 2008; Gneiting and Raftery, 2007), to which
the energy score reduces in dimension $d = 1$. Based on $ES$, the energy skill score (ESS) is defined as

$$\text{ESS} = 1 - \frac{\overline{\text{ES}}_{\text{projection}}}{\overline{\text{ES}}_{\text{reference}}} \tag{24}$$

where $\overline{\text{ES}}_{\text{projection}}$ is the mean ES of the projection model analyzed and $\overline{\text{ES}}_{\text{reference}}$ is the mean ES of the reference projec-
tion. The ESS ranges from $-\infty$ to 1. ESS < 0 indicate that the reference projection has higher skill than the projection model,
ESS = 0 implies equal skill, and ESS > 0 means that the projection model has a higher skill, with ESS = 1 representing a perfect
score. For this study, we considered the reference projection (climatology) as a benchmark. ESS was computed for both the
calibration and validation models.

## 4 Results

### 4.1 Selection of the best model for each lead time

For each lead time, different candidate BHMs were calibrated for the period 1965-2018, and the best BHM was selected based
on the lowest DIC value.





**Table 2.** DIC values for different candidate BHMs for a 0-month lead time and the best model for the other lead times. For each model, the same covariates are considered at all gauges for the location parameter. Candidate BHMs are sorted from the lowest to the highest DIC value for a 0-month lead time. Scale and shape parameters are considered stationary. All candidate BHMs consider a Gaussian copula to model spatial dependence.

| Lead time | Model | Covariates | DIC |
|---|---|---|---|
| | Nonstationary | SASWE, SAAMT | 1021.5 |
| | Nonstationary | SASWE, AMO, SAAMT | 1030.9 |
| | Nonstationary | SASWE, ENSO, SAAMT | 1032.7 |
| | Nonstationary | SASWE, PDO, SAAMT | 1032.7 |
| | Nonstationary | SASWE | 1034.5 |
| 0-month | Nonstationary | SASWE, ENSO, AMO | 1045.3 |
| | Nonstationary | SASWE, PDO, AMO | 1047.6 |
| | Nonstationary | SASWE, ENSO, PDO | 1049.7 |
| | Nonstationary | SASWE, ENSO | 1077.8 |
| | Stationary | – | 1132.8 |
| | Nonstationary | SASWE, AMO | 1151.4 |
| | Nonstationary | SASWE, PDO | 1173 |
| 1-month | Nonstationary | SASWE, PDO | 1065.2 |
| 2-months | Nonstationary | SASWE, ENSO | 1075.6 |

Table 2 shows DIC values for different candidate BHMs for a 0-month lead time and the best model for the other lead times (last two rows). In the case of a 0-month lead time, the best model corresponds to the first row of Table 2. Notice that for a 0-month lead time predictive skill resides in SWE and air temperature as this hydroclimatic information relates well to the streamflow extremes in May-June (see Fig. 5). However, for longer lead times of 1 and 2-months, the large-scale climate indices add value. All the models fitted consider a Gaussian copula to model spatial dependence. Detailed tables with different models fitted for a 1-month and 2-months lead time can be found in Tables A1 and A2.

### 4.2 Ability of the BHM to capture spatio-temporal dependence

To assess the ability of the BHM to capture spatio-temporal dependence through nonstationary covariates and a Gaussian copula, we compared the best model for the 0-month lead time selected in the previous section (i.e., SASWE and SAAMT as covariates with a Gaussian copula) against three models that do not consider a Gaussian copula: Stationary, nonstationary with SASWE as covariate, and nonstationary with SASWE and SAAMT as covariates.

Figure 7 shows the Energy Skill Score (ESS) distribution from the calibration (1965-2018) for the four BHMs for a 0-month lead time. Higher values of the ESS are better. The stationary model (light grey box) performs poorly in capturing the spatio-temporal dependence compared to the benchmark (i.e., values below 0). This model also shows a lower variability of the skill





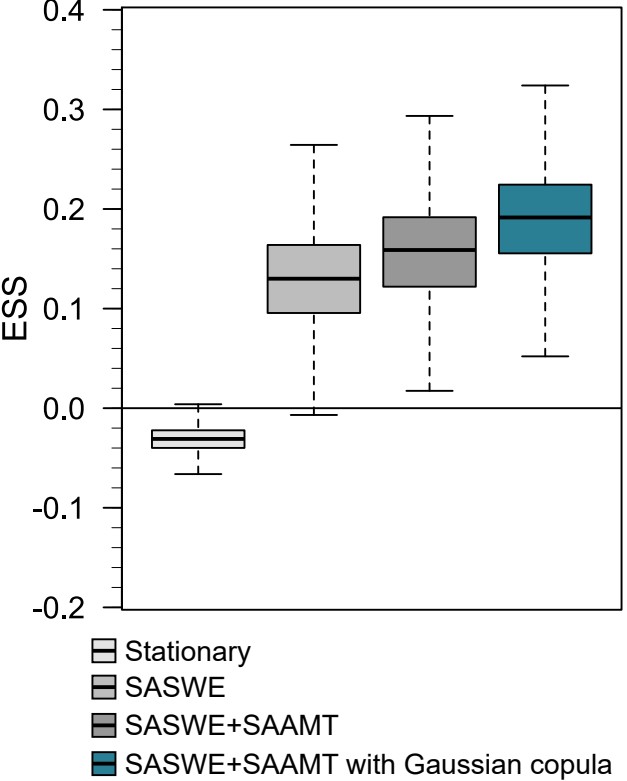

**Figure 7.** Energy Skill Score (ESS) distribution for four different BHM versions for a 0-month lead time from the calibration period (1964-2018). Higher values of the ESS are better. The whiskers show the 95% credible intervals, boxes the interquartile range, and horizontal lines inside the boxes, the median. Outliers are not displayed. The light grey box denotes a stationary model, the grey box a nonstationary model with SASWE as a covariate, the dark grey box a nonstationary model with SASWE and SAAMT as covariates, and the dark turquoise model, adds a Gaussian copula to model spatial dependence.

score than the other Bayesian hierarchical models, which is expected because the stationary model produces the same projection for each year. When a nonstationary model with SASWE as a covariate is considered (grey box), the skill is substantially improved compared to the benchmark and the stationary model (i.e., most of the values above 0). This improvement is consistent with Fig. 5 and previous studies stating that SWE until the beginning of the spring season is the most skillful predictor of

spring-summer seasonal streamflow in mountainous regions (Koster et al., 2010; Livneh and Badger, 2020). When SAAMT is added as a covariate (dark grey box), there is a significant improvement of ESS compared to the model with only SASWE. The best model with copula (dark turquoise box) substantially increases model skill. Model skill increases were tested with a 95 % confidence interval using a nonparametric test for the median (Gibbons and Chakraborti, 1992).

     To further highlight the ability of the Gaussian copula to capture spatio-temporal dependence, we computed spatial joint

threshold non-exceedance probabilities of the seven stations and compared them to non-exceedance probabilities derived from three models without the copula, i.e., spatially independent models for each streamflow gauge. In Figure 8, we display the

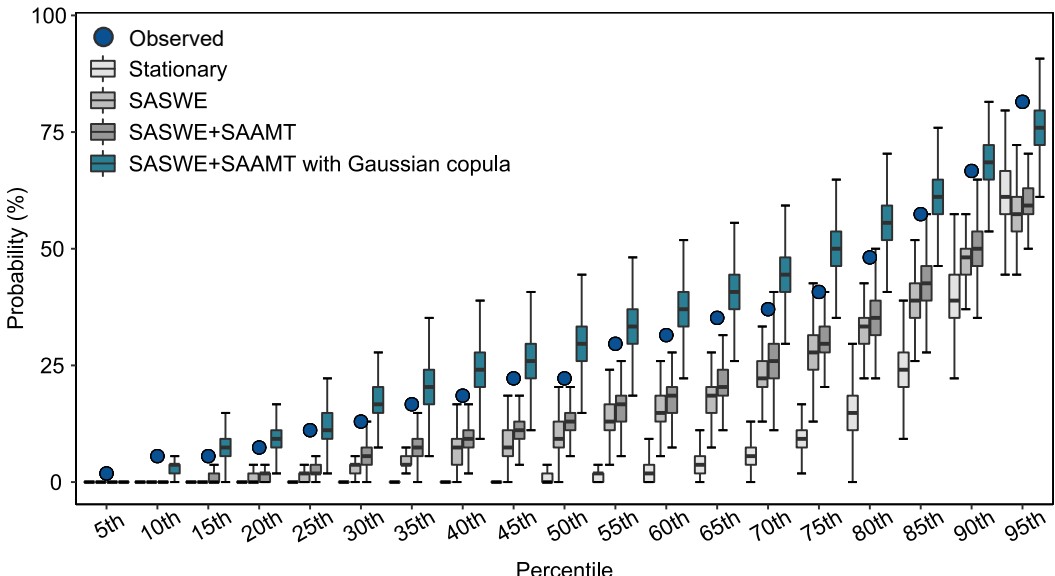

**Figure 8.** Distribution of the probability that all gauges do not exceed their $k$th percentile for four different versions of the BHM for a 0-month lead time for the calibration period (1964-2018). The whiskers show the 95% credible intervals, boxes the interquartile range, and horizontal lines inside the boxes, the median. Outliers are not displayed. The observations are shown as blue circles and the same color scheme as in Fig. 7 was considered for the four versions of the BHM.

distribution of the probability that all streamflow gauges do not exceed their $k$th percentile for the observations and the four models shown in Fig. 7 for a 0-month lead time and the calibration period. The model with nonstationary covariates and a Gaussian copula (dark turquoise box plots) can almost reproduce the shape of the observed distribution in terms of the interquartile range (boxes), which is an indication that this model can capture the spatio-temporal dependence of the data well. Figure 8 also shows that even without considering a copula, a stationary model with skillful covariates can partially capture the spatio-temporal dependence (see in Fig. 8 the difference of the distribution shape for the stationary and nonstationary models without copula). This suggests that with increasing skill of the covariates, the added value of the copula gets smaller. Similar performance was observed for the nonstationary model with SASWE as a covariate plus a Gaussian copula and for a nonstationary model with a less skillful covariate (ENSO) plus a Gaussian copula (not shown here).

In order to assess the spatial performance of the BHM, Figure 9 shows the time series of the spatial average projection of spring season maximum specific streamflow over all seven gauges of the UCRB from the calibration (1964-2018) for the best model for a 0-month lead time without a Gaussian copula and with a Gaussian copula. Our results show that by adding a Gaussian copula to the BHM (Fig. 9b), observations can be better captured by the ensembles' median; and the ensembles represent a higher variability, which allows capturing some observations that are not captured without the copula (Fig. 9a).



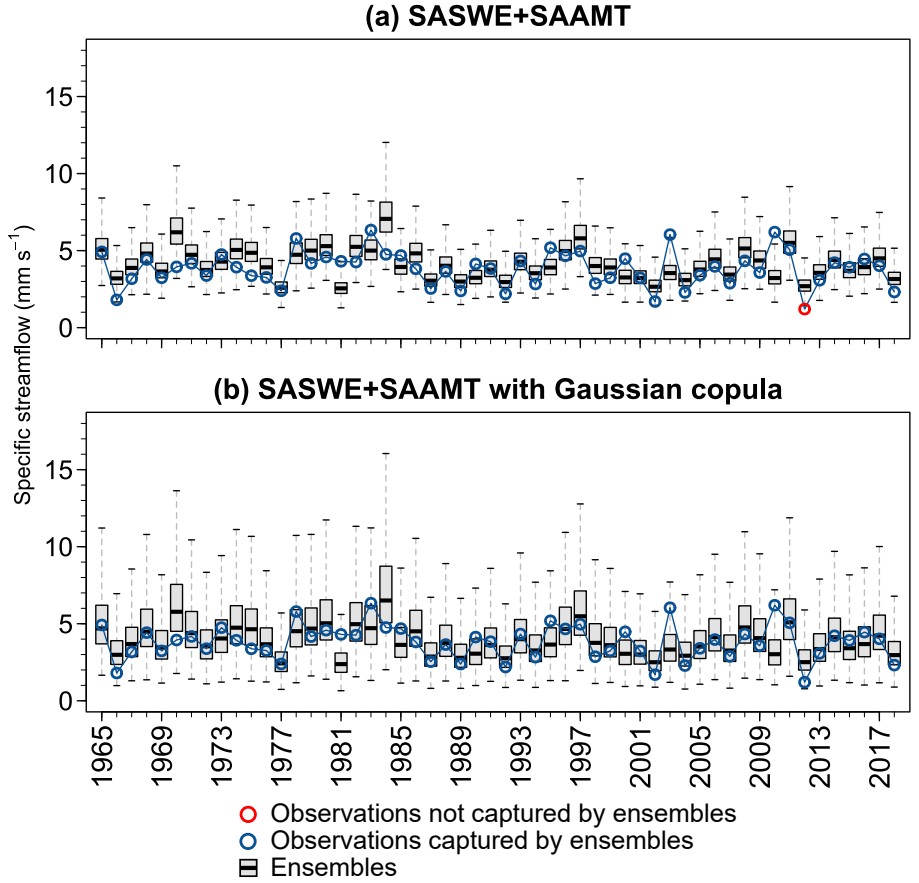

**Figure 9.** Time series of average projected maximum specific spring flow over all seven gauges of the UCRB $(\mathrm{mm\,d^{-1}})$ from the calibration (1964-2018) for the best model for a 0-month lead time (a) without a Gaussian copula and (b) with a Gaussian copula. Blue and red points indicate observations captured or not by the ensemble's variability, respectively. Whiskers indicate the 95% credible intervals, boxes the interquartile range, and horizontal lines inside the boxes, the median. Outliers are not displayed.

## 4.3    Model performance at different lead times

To define the extent to which seasonal streamflow extremes projections for longer lead times can be skillful, we assess the performance of the BHM at different lead times for the calibration and leave-1-year-out cross-validation, including cross-validation focusing on extremes (60th percentile). Covariates considered for each lead time were presented in section 4.1.

Figure 10 displays the Energy Skill Score (ESS) distribution for different lead times from calibration, leave-1-year-out cross-validation, and leave-1-year-out cross-validation on extremes. EES decreases as the lead time increases (Fig. 10a). However, the three lead times capture the multivariate dependence better than the calibration benchmark (ESS values above 0). For the leave-1-year-out cross-validation (Fig. 10b), there is a substantial decrease in performance compared to the calibration, except for the 0-month lead time (dark turquoise boxplot). However, model performance for the 1-month lead time is better than the





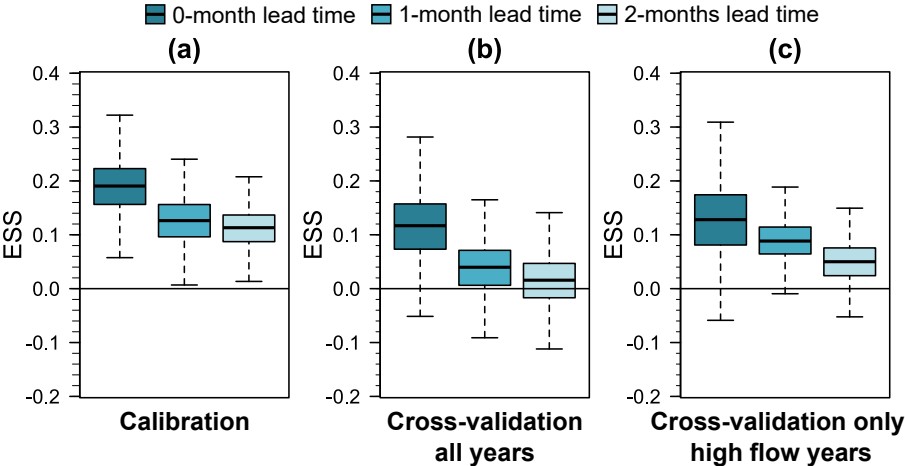

**Figure 10.** Energy Skill Score (ESS) distribution for different lead times from (a) calibration; (b) leave-1-year-out cross-validation; and (c) leave-1-year-out cross-validation for extremes (60th percentile). Dark turquoise boxplots denote a 0-month lead time, turquoise boxplots a 1-month lead time, and light turquoise boxplots a 2-months lead time. Higher values of the ESS indicate better model performance. Whiskers indicate the 95% credible intervals, boxes the inter-quartile range, and the horizontal lines inside the boxes, the median. Outliers are not displayed. All the models consider a Gaussian copula.

benchmark (ESS quartiles above 0), and the model for a 2-months lead time shows lower performance than the benchmark only for the third quartile (above 0). Finally, for the leave-1-year-out validation for extremes (Fig. 10c), all lead times show higher skill than the benchmark (ESS quartiles above 0), highlighting the potential usefulness of the model for flood risk assessments.

To assess at site performance, we computed the continuous rank probability skill score (CRPS; Gneiting and Raftery, 2007; Hersbach, 2000) at each gauge (Fig. A4). As for the ESS, the CRPSS ranges from $-\infty$ to 1, and its values have the same mean-
ing. We obtained similar results to ESS (Fig. 10) with the exception of a few gauges for leave-1-year-out cross-validation and leave-1-year-out cross-validation for extreme where the performance was poorer than the benchmark for the cross-validation.

Figure 11 shows the time series of average projected maximum specific spring flow over all seven gauges of the UCRB ensembles and the distributions of the Pearson correlation coefficient from the leave-1-year-out cross-validation for the three lead times and the benchmark. Simulations relying on models with a copula show a similar variability and can capture observed
values inside of their ensembles' variability for all three lead times (Fig. 11a-b). The benchmark (Fig. 11d) cannot capture the observations since it converges to a stationary model, i.e., it gives the same projection for all years. There is a slight performance reduction for models with 1- and 2-months lead times compared to the 0-month lead time (Fig. 11e). The medians of the Pearson correlation coefficient for the three lead times vary between 0.37 and 0.5 while the median for the benchmark is close to -1. In addition, model performance for the calibration (Fig. A5) is similar to the one of the cross-validation. This result
indicates only small performance reductions for projections and implies that the framework proposed could be useful for the early implementation of flood risk adaptation strategies each year.

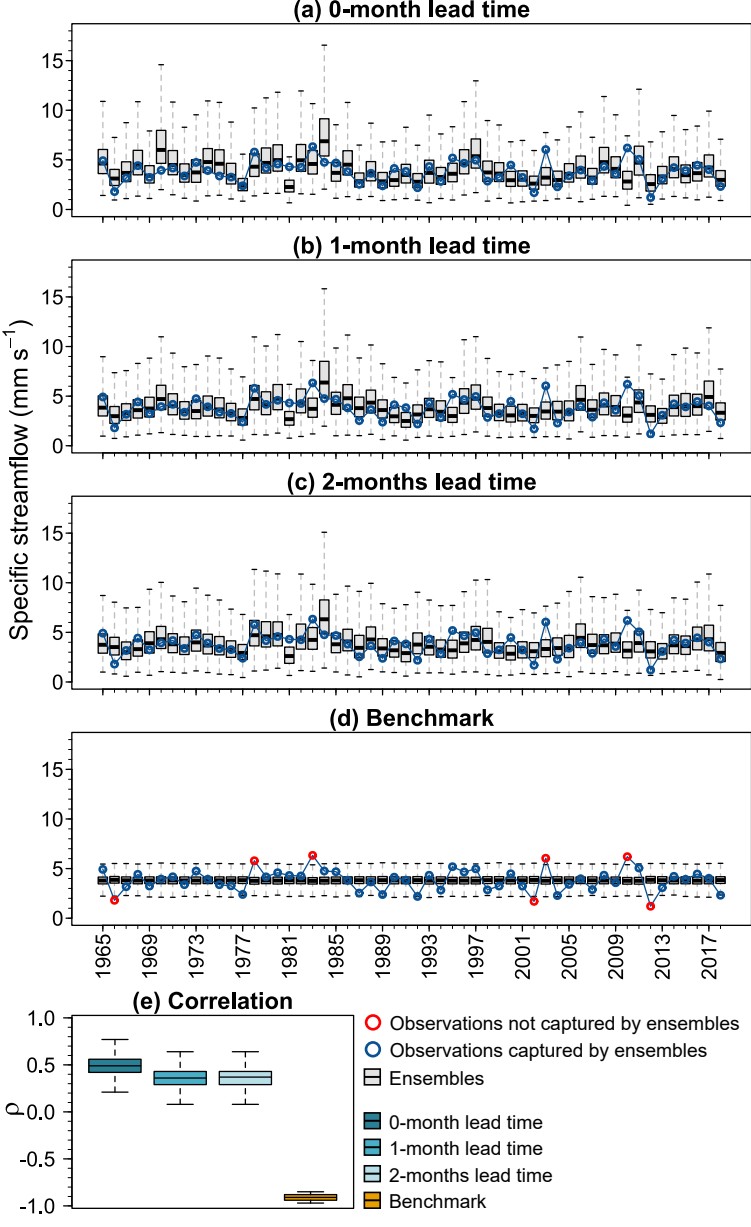

**Figure 11.** Time series of average projected maximum specific spring flow over all seven gauges of the UCRB, $(mm\,d^{-1})$, from the leave-1-year-out cross-validation for a (a) 0-month lead time; (b) 1-month lead time; (c) 2-months lead time; (d) benchmark; and (e) distributions of the Pearson correlation coefficient between observed and ensembles of average maximum specific spring flow over all seven gauges for the different models. Blue and red points in panels (a) – (d) indicate when observations are captured or not by the ensembles' variability, respectively. Whiskers show the 95% credible intervals, boxes the interquartile range, and horizontal lines inside the boxes, the median. Outliers are not displayed. All lead time models consider a Gaussian copula.





## 5  Discussion

Compared to operational forecast models that consider short lead times and seasonal streamflow forecast models that are useful for reservoir operation with a focus on water availability during the dry season, the proposed BHM proposed here has
the following benefits:

- Allows for considering potential climate change effects by modeling the margins in a nonstationary setting using suitable covariates.

- Allows for capturing spatio-temporal dependence by including a Gaussian copula. Consequently, the spatial BHM captures observations that are not captured by the average projection of maximum specific spring flow of a BHM without
copula

- Provides joint projections of maximum spring flow up to 2-months in advance by relying on the skill of snow accumulated during the winter season.

A question coming to mind is: how can the proposed modeling framework be used to deliver interpretable seasonal projections? It might be difficult for decision-makers to make decisions based on average maximum specific spring flow over all
seven gauges of the UCRB (see Fig. 11). To overcome this difficulty, we propose to provide joint projections of maximum spring flow by providing the first three quartiles of the ensembles along with some past streamflow values as a reference. Providing reference values can help to make decisions about risk adaptation up to two months in advance. Reference values can be, e.g., the observed median specific streamflow for an average year without flood occurrence; the maximum observed specific streamflow on the record; or the lowest observed specific streamflow (threshold) that can cause flood occurrence. To
find the threshold flow that can cause flood occurrence, we computed the ratio between average maximum and average mean specific spring flow over all seven gauges of the UCRB. Then, we picked the average maximum specific spring flow value of the year with the highest ratio value, which corresponds to the threshold flow. The reason for this is that when the basin is drier (high ratio between peak runoff and total seasonal runoff), even if the peak streamflow is not so high, a flood can occur. Based on the lowest observed specific streamflow (threshold, $q_{\text{thresh}}$) that can trigger flood occurrence we define a potential
flooding alarm system. This system defines different flooding alarm levels by comparing the threshold against the first three quartiles (exceedance probability, $q_{25\text{th}}$, $q_{50\text{th}}$, and $q_{75\text{th}}$) of joint projections of maximum spring flow for the year analyzed. This potential flooding alarm system is shown in Fig. 12a. Thus, for each lead time, the flooding alarm is activated with a low risk of flooding if $q_{25\text{th}} > q_{\text{thresh}}$; moderate risk of flooding if $q_{50\text{th}} > q_{\text{thresh}}$; or high risk of flooding if $q_{25\text{th}} > q_{\text{thresh}}$. In all other cases, the alarm is not activated.
To illustrate this system, Figure 12b-c show examples of when the potential flooding alarm is successfully activated and not activated. Figure 12b presents the joint projections of maximum spring flow of the UCRB for 2011 at three lead times (0-2 months), which were obtained from the leave-1-year-out cross-validation, along with the three reference values mentioned above and the observed value for 2011. It can be seen that based on the projections, The flooding alarm is activated with a low risk of flooding by March 1st (2-months lead time), moderate risk of flooding by April 1st (1-month lead time), and





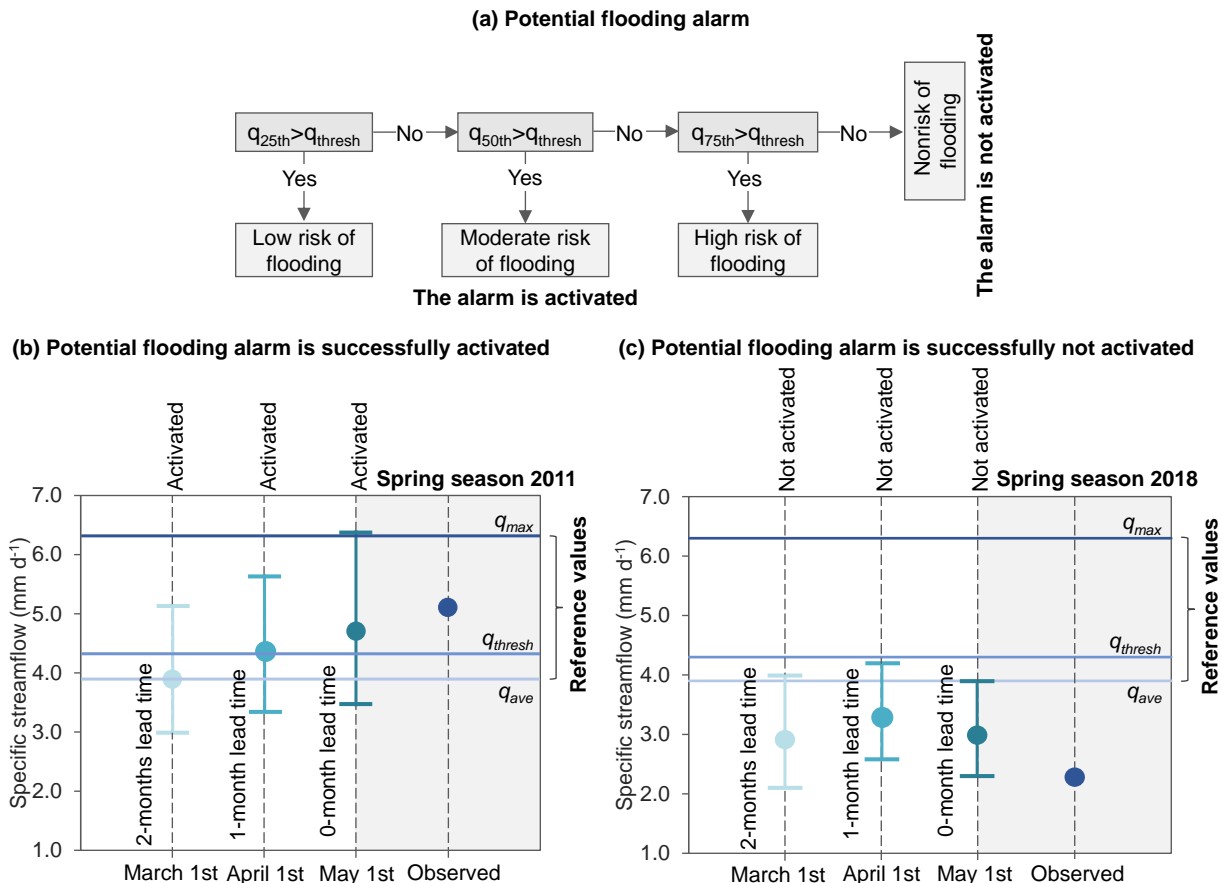

**Figure 12.** (a) Schematic of the potential flooding alarm system proposed. The flooding alarm is activated if $q_{25th} > q_{thresh}$ (low risk of flooding), $q_{50th} > q_{thresh}$ (moderate risk of flooding), or $q_{75th} > q_{thresh}$ (high risk of flooding). Otherwise, the alarm is not activated. (b) joint projections of spring maximum streamflow of UCRB for 2011 when flooding is successfully activated, and (c) joint projections of spring maximum streamflow of UCRB for 2018 when flooding is successfully not activated. Projections at a 0-, 1-, and 2-months lead times correspond to dark turquoise, turquoise, and light turquoise, respectively. The blue point corresponds to the joint observed flow for 2018, horizontal lines correspond to the joint observed highest flow ($q_{max}$, dark blue), joint observed flooding threshold flow ($q_{thresh}$, blue), and joint observed average flow ($q_{ave}$, light blue). For each lead time, the whiskers show the first and third quartile ($q_{25th}$ and $q_{75th}$), the point, the median or second quartile ($q_{50th}$).

high risk of flooding by May 1st (0-month lead time). Thus, the flooding alarm is successfully activated before spring since the observed value for 2011 exceeded the threshold flow, and actually, flood impacts were documented in 2011 (Werner and Yeager, 2013). In addition, Figure 12c shows the projection of maximum spring flow of the UCRB for 2018 when the flooding alarm is successfully not activated since the three quartiles for each lead time and the observed flow for 2018 are below the threshold flow.





Model performance could be further improved by including additional potentially skillful covariates such as specific ocean and atmosphere features that are expected to have a stronger relationship with the basin streamflow and extremes (e.g.; Bracken et al., 2010; Grantz et al., 2005; Regonda et al., 2006). The framework can be applied to another region by adjusting the choice of covariates to covariates suitable for the region of interest. In addition, it can be easily adjusted such that it represents future climate conditions if future projections of the covariates are available. In the application presented here, we only modeled the
location parameter as nonstationary. If the framework is applied to another basin, this modeling choice has to be reconsidered. It is advisable as a first step to do an initial run of the model for defining which parameters should be considered nonstationary. Although an advantage of copulas is that the dependence structure can be estimated independent of the margins, future model development may also explore the connection of copula and margins fitting during the MCMC simulations. Currently, the BHM uses pseudo-observations to estimate the copula dependence parameters (Eq. (8); Bracken et al., 2018; Brunner et al.,
2019; Genest and Favre, 2007).

## 6 Summary and conclusions

In this study, we presented a Bayesian Hierarchical Model (BHM) to project seasonal streamflow extremes for several catchments in a river basin for several lead times. The streamflow extremes at a number of gauge locations in a basin are modeled using a Gaussian elliptical copula and Generalized Extreme Value (GEV) margins with nonstationary parameters. These pa-
rameters are modeled as a linear function of suitable covariates from the previous season.

We applied this framework to project 3-day maximum spring (May-June) flow at seven gauges in the Upper Colorado River Basin (UCRB) network, at 0-, 1-, and 2-months lead time. As potential covariates, we used indices of large-scale climate teleconnections – ENSO, AMO, and PDO, regional mean snow water equivalent, and temperature from the preceding winter season.

From the analysis of different models for a 0-month lead time, we can conclude that:

- Spatial average snow water equivalent (SASWE) accumulated during fall and spring is the most skillful predictor of spring season maximum streamflow across the UCRB.

- The increase in BHM performance is low when adding other climatic indices such as PDO.

- Including a copula in the BHM enables capturing the spatio-temporal dependence of streamflow extremes which is not
fully possible with independent marginal models.

The comparative analysis for three different lead times revealed that increasing the lead time from 0 to 2 months only weakly decreases model skill. This finding implies that the framework proposed could be useful for the early implementation of flood risk adaptation and preparedness strategies. We propose to guide decision making by providing joint projections of maximum spring flow as the first three quartiles of the ensembles of the spatial average projection of spring season maximum specific
streamflow over all seven gauges of the UCRB along with past reference streamflow values. Such a communication strategy could help decision-makers to implement adaptation strategies that address the spatial dimension of flooding.





## Appendix A:  Supplementary information

**Table A1.** DIC values for different candidate BHMs for a 1-month lead time. For each model, the same covariates are considered at all gauges for the location parameter. Candidate BHMs are sorted from the lowest to the highest DIC value. Scale and shape parameters are considered stationary. All candidate BHMs consider a Gaussian copula to model spatial dependence.

| Model | Covariates | DIC |
| --- | --- | --- |
| Nonstationary | SASWE, PDO | 1065.2 |
| Nonstationary | SASWE, ENSO | 1067.3 |
| Nonstationary | SASWE | 1069.7 |
| Nonstationary | SASWE, AMO | 1071.9 |
| Nonstationary | SASWE, ENSO, PDO | 1077.4 |
| Nonstationary | SASWE, PDO, AMO | 1080.6 |
| Nonstationary | SASWE, ENSO, AMO | 1117.1 |
| Stationary | – | 1132.8 |

**Table A2.** DIC values for different candidate BHMs for a 2-months lead time. For each model, the same covariates are considered at all gauges for the location parameter. Candidate BHMs are sorted from the lowest to the highest DIC value. Scale and shape parameters are considered stationary. All candidate BHMs consider a Gaussian copula to model spatial dependence.

| Model | Covariates | DIC |
| --- | --- | --- |
| Nonstationary | SASWE, ENSO | 1075.6 |
| Nonstationary | SASWE | 1078.6 |
| Nonstationary | SASWE, PDO | 1090.8 |
| Nonstationary | SASWE, PDO, AMO | 1097.6 |
| Nonstationary | SASWE, ENSO, PDO | 1099.1 |
| Stationary | – | 1132.8 |
| Nonstationary | SASWE, AMO | 1144.9 |
| Nonstationary | SASWE, ENSO, AMO | 1158.5 |

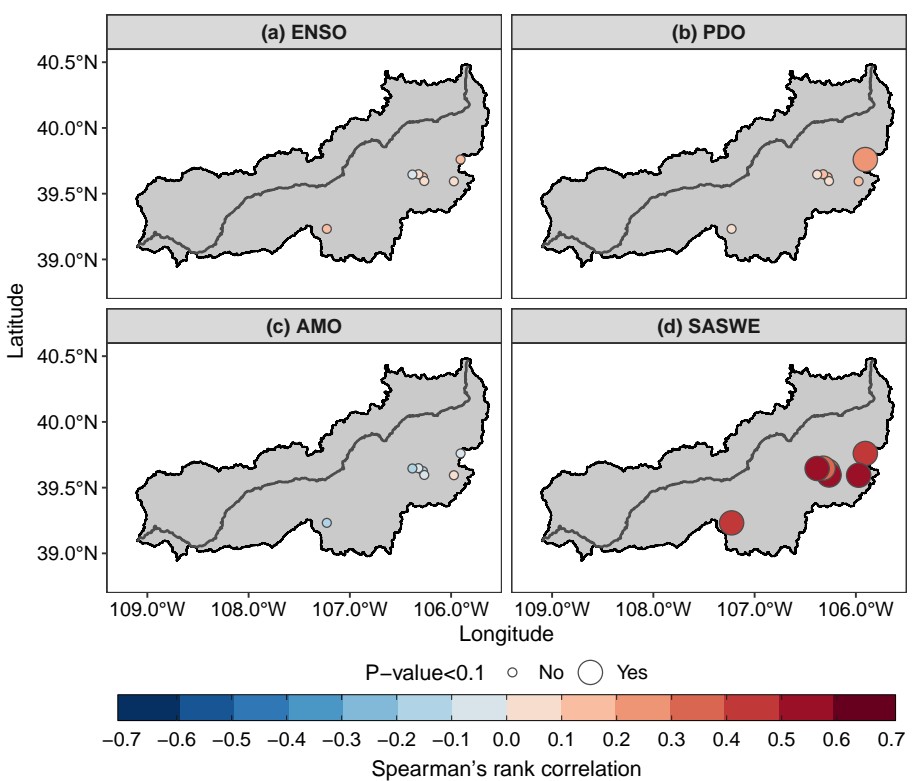

**Figure A1.** Spearman's rank correlation coefficient between spring 3-day maximum streamflow and potential covariates for a 1-month lead time: (a) Mean Nov-March ENSO; (b) Mean Nov-March PDO; (c) Mean Nov-March AMO; and (d) the spatial average of Nov-March snow water equivalent (SASWE). Big circles indicate that Spearman's rank correlation is significant (P-value<0.1).



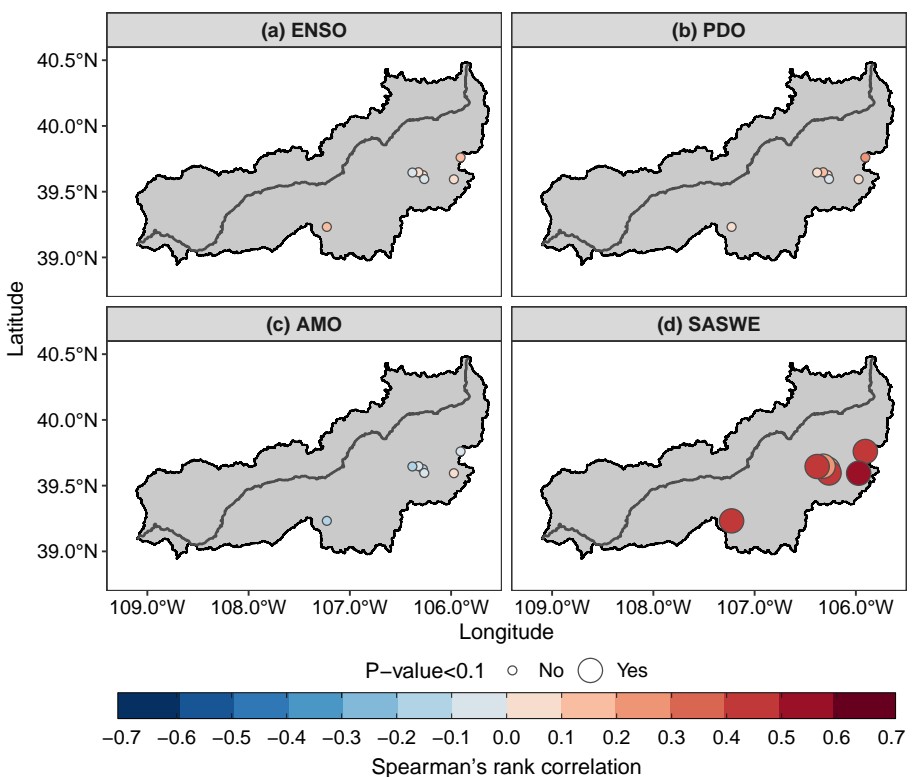

**Figure A2.** Spearman's rank correlation coefficient between spring 3-day maximum streamflow and potential covariates for a 2-months lead time: (a) Mean Nov-Feb ENSO; (b) Mean Nov- Feb PDO; (c) Mean Nov- Feb AMO; and (d) the spatial average of Nov- Feb snow water equivalent (SASWE). Big circles indicate that Spearman's rank correlation is significant (P-value<0.1).



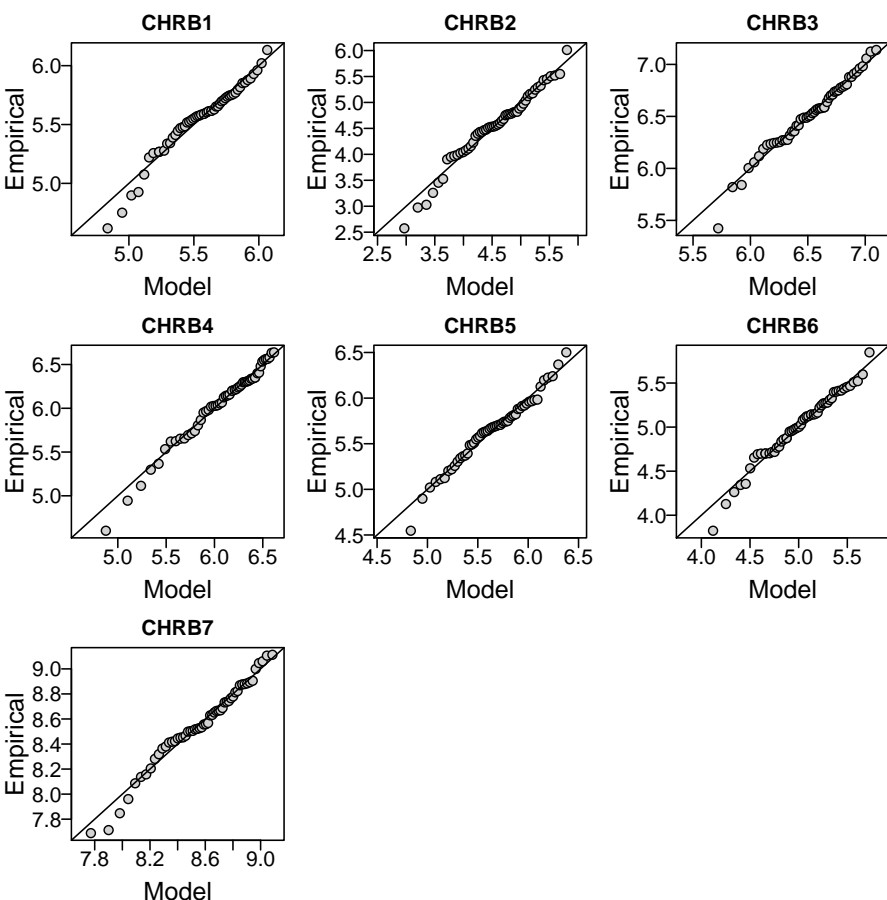

**Figure A3.** Q-Q plots of the stationary GEV distribution fitted for spring 3-day maximum streamflow for seven streamflow gauges of UCRB.

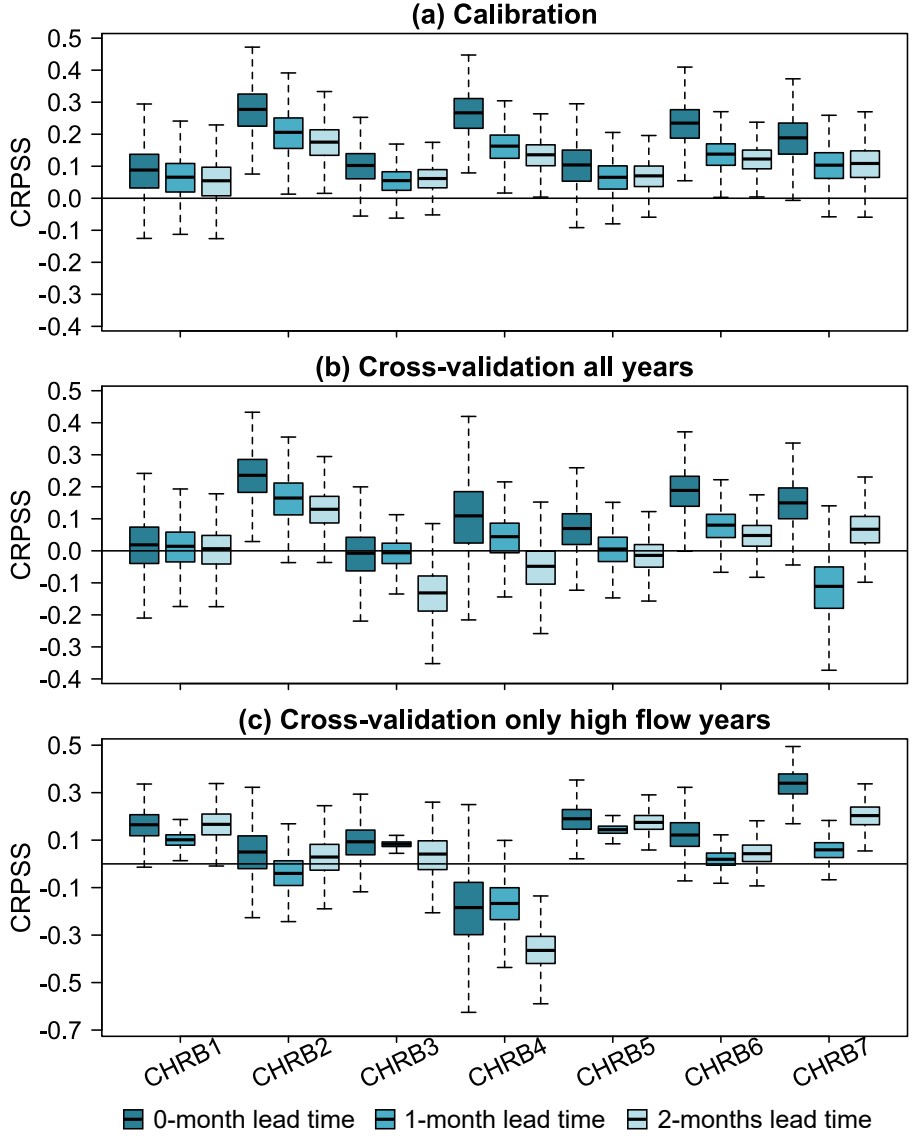

**Figure A4.** The continuous rank probability skill score (CRPS) distribution for different lead times from (a) calibration; (b) leave-1-year-out cross-validation; and (c) leave-1-year-out for extremes (60th percentile) cross-validation. Dark turquoise boxplots denote a 0-month lead time, turquoise boxplots a 1-month lead time, and light turquoise boxplots a 2-months lead time. Higher values of the ESS are better. The whiskers show the 95 % credible intervals, boxes the interquartile range, and the horizontal lines inside the boxes, the median. Outliers are not displayed. All the lead time models consider a Gaussian copula. As for the ESS, the CRPSS ranges from -∞ to 1, and its values have the same meaning.

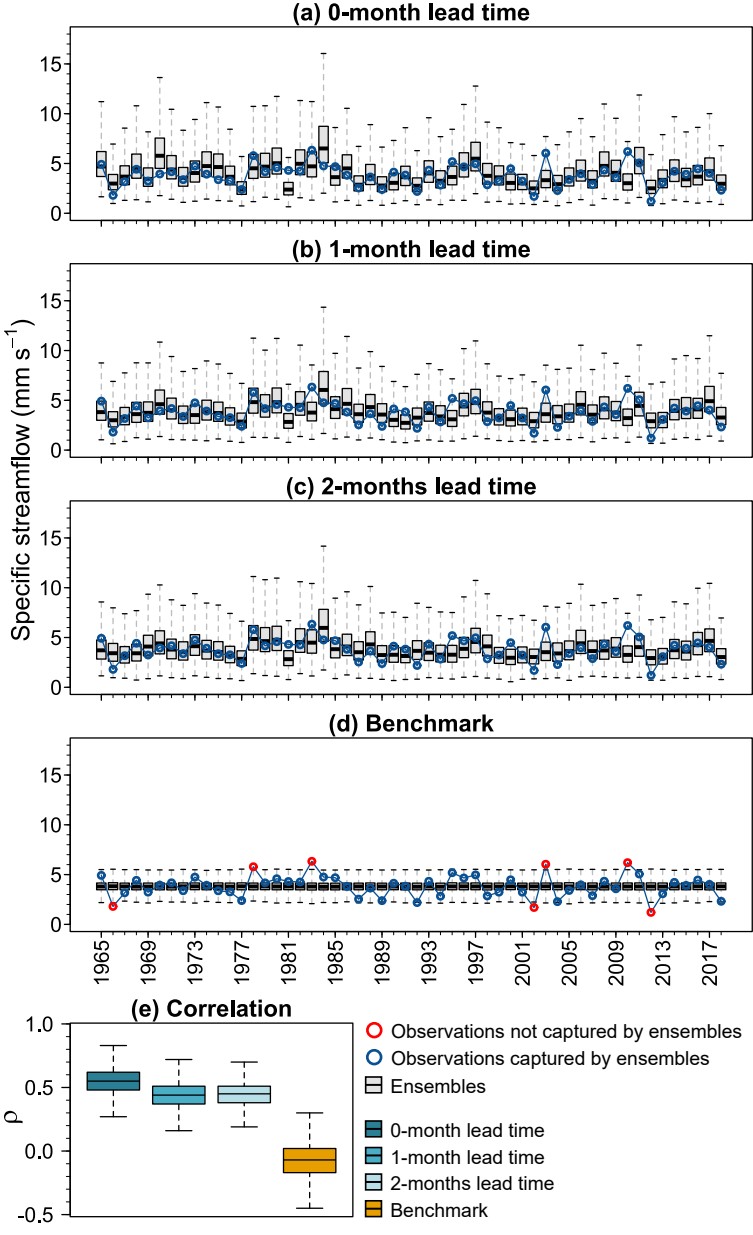

**Figure A5.** Time series of the average projection of maximum spring flow over all seven gauges of the UCRB, in mm d$^{-1}$, from the calibration for (a) 0-month lead time; (b) 1-month lead time; (c) 2-months lead time; (d) benchmark; and (e) distributions of the Pearson correlation coefficient between observed and ensembles of average spring maximum specific streamflow over all seven gauges for the different models. Blue and red points in panels (a) – (d) indicate when observations are captured or not by the ensembles' variability, respectively. The whiskers show the 95 % credible intervals, boxes the interquartile range, and horizontal lines inside the boxes, the median. Outliers are not displayed. All lead time models consider a Gaussian copula.



*Data availability.* The dataset used in this study, which consists of time series of potential covariates and 3-day maximum spring streamflow for the seven station gauges, can be downloaded from HydroShare https://doi.org/10.4211/hs.d8c1b413951843cf9be968e9d2a4aa79

*Author contributions.* The idea and setup for the paper were jointly developed by the four co-authors. The model implementation and analysis were performed by AO and discussed with the co-authors. AO wrote the first draft of the manuscript which was revised and edited by MIB, RB, and WK

*Competing interests.* The authors declare that they have no conflict of interest.

*Acknowledgements.* This project was funded by the National Science Foundation grant 1243270. We also acknowledge the support from
the Fulbright Foreign Student Program and the Comisión Nacional de Investigación Científica y Tecnológica (CONICYT) Scholarship Program/DOCTORADO BECAS CHILE/2015-56150013 for the first author. The second author was supported by the Swiss National Science Foundation via a PostDoc.Mobility grant (Number: P400P2_183844). Partial support from the Monsoon Mission Project of the Ministry of Earth Sciences, India, for first and second authors is thankfully acknowledged. The fourth author was supported by NSF DMS-1811294 and DMS-1923062.





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
