# Peer review of "A space-time Bayesian hierarchical modeling framework for projection of seasonal maximum streamflow"

_Hydrology and Earth System Sciences, 2021_

## Author Comment (AC1)

Response to the Referee #1

Thank you for your comments and valuable suggestions. Below are our point-by-point responses to the referee's comments (in italic). We hope they will find them to be comprehensive and satisfactory. In the responses, we refer to specific figures or lines in the main text, to allow the referee to follow the changes implemented in the revised manuscript.

*The manuscript "A space-time Bayesian hierarchical modeling framework for projection of seasonal streamflow extremes" by Ossandón et al. proposes a Bayesian Hierarchical Model (BHM) to project seasonal streamflow extremes for multiple catchments in a river basin up to 2 months lead time. The spatio-temporal dependence is modelled through a Gaussian elliptical copula and Generalized Extreme Value margins with nonstationary parameters and covariates. The proposed model is used to model streamflow extremes at 7 gauges location in the Upper Colorado River Basin (UCRB). The proposed framework and its application to the UCRB are interesting and well presented. I have some (minor) comments, especially concerning the application of the model to the UCRB.*
**Response:** Thank you very much for acknowledging the value of our work and for the constructive feedback.

1. *Choice of the timescale of the indicator for seasonal streamflow extremes: why are 3-day maxima considered, and not, e.g. 1-day maxima or instantaneous (seasonal) peaks? Especially for small and mountainous catchments (catchment area ~10-20 km$^2$, as the 7 considered in this application) the 3-day maximum discharges are not really representative for flood events because of the fast response times of the catchments and the consequent high variability of the discharge in short periods of time. In such small catchments, maximum discharges at smaller timescales (e.g. instantaneous peak discharges) are better representatives for flood dynamics and magnitude of the peaks and, therefore, better indicators for flood risk management strategies.*

   **Response:**
   We agree with the reviewer that small and mountainous catchments can have fast response times, particularly when events are rainfall driven, and 1-day maxima may be used to describe some of the flood events caused by intense rainfall. However, the streamflow regime of the catchments considered is snowmelt-dominated. Therefore, our analysis focuses on snowmelt-driven floods in late spring (May-June), which are the dominant flood season. As snowmelt-driven events typically have longer durations and flood volumes than events driven by convective storms in summer, we considered 3-day maxima instead of 1-day maxima. Still, we want to stress that the framework proposed can also be applied to 1-day maxima, which we specified in the Discussion section. **See lines 426-427** in the revised manuscript.
   We also add a statement to the introduction. **See lines 66-67** in the revised manuscript.

2. *Selection and location of the sites: sites are selected for the application according to the length of the streamflow series available. The selection results in 7 sites (fig 2), mostly located very close to each other and not well distributed across the UCRB. Are there nested catchments? What are the implications of this uneven spatial distribution on application results?*

   **Response:**
   The seven catchments are not nested; they are individual sub-catchments of the larger Upper Colorado River Basin (UCRB). The uneven spatial distribution is accounted for through the Gaussian copula's dependence matrix, which can represent both weak and high correlations.

The sites used for the application in our manuscript are spatially correlated, which is shown in Figure 3 of the revised manuscript. The correlation also becomes apparent in Figure 10 (revised manuscript), where one can see that considering a dependence structure (Gaussian copula) in the hierarchical model allows us to capture the observed values of maximum specific spring flow over all seven gauges of the UCRB (Figure 10b), which is not possible by making projections from a model that only considers the marginal distributions.

We addressed the nestedness aspect to the streamflow data section. Please, **see line 186** in the revised manuscript.

3. *Choice of covariates: the authors state that :" In this basin, almost all extremes that cause severe flooding occur in spring as a result of snowmelt and precipitation" (lines 11-12) and "Floods are a concern in mountain regions such as the Upper Colorado River Basin (UCRB), where streamflow extremes happen in spring due to snowmelt in combination with precipitation and are projected to increase under future climate conditions" (lines 23-25). However, only (regional) covariates related to snowmelt processes are considered (SASWE and SAAMT). Why is measured precipitation not accounted for in this application? Why only considering the temperature in April and not, e.g. the spring mean or max temperature? Furthermore, since the role of non-stationarity is emphasised in the paper, I am wondering whether the relevance of these 2 processes (snowmelt and precipitation) in causing flood events is unchanged during the observation period and if it is expected to be the same in the future, or if other processes are becoming more relevant, as observed in other parts of the world (e.g. summer floods due to extreme precipitation becoming more relevant in areas that used to be snow-dominated due to a warmer climate).*

**Response:**

We considered regional covariates related to snowmelt processes because they are good seasonal predictors of spring flow well ahead of time (Koster et al., 2010; Livneh & Badger, 2020). In contrast to snow-related variables, precipitation can not be well predicted ahead of more than two weeks (Werner & Yeager, 2013). As this paper aims to **provide seasonal projections for lead times between 0-2 months** (released on May 1st – March 1st), we did not consider precipitation covariates. For the same reason, we did neither consider the spring mean or max temperature. We clarified this in the covariates section of the revised manuscript. **See lines 213-214** of the revised manuscript.

To provide seasonal projections, non-stationarity is considered (we want to capture the temporal variability). Otherwise, we will get the same projection each year. To illustrate this, we show the projections of the stationary model below (Figure 1 in the responses to the reviewer). As can be seen in the figure, the stationary model provides the same projection for each year since the simulations come from the same distribution for all the years. That does not happen when one considers a nonstationary model (Figure 9 of the manuscript and Figure 10 of the revised manuscript).

[Figure]

**Figure 1:** Time series of average projected spring maximum specific streamflow over all seven gauges of the UCRB (mm d$^{-1}$) from the calibration (1965-2018) for the stationary BHM for a 0-month lead time. Blue and red points indicate observations captured or not by the ensemble's variability, respectively. Whiskers indicate the 95% credible intervals, boxes the interquartile range, and horizontal lines inside the boxes, the median. Outliers are not displayed. Ensemble refers to the set of projections produced for each year.

In addition, we agree with the reviewer that the relative importance of the two processes snowmelt and precipitation in causing flood events may change in the future, with precipitation becoming relatively more important. However, In the case of headwater basins in mountain regions such as the one considered in this study, snowmelt will remain the dominant flood generation process in the future, as shown by climate change projections in the region (Safeeq et al., 2016). Consequently, its predictive skill might slightly decrease.

Finally, we would like to point out that we are proposing a framework that can be adjusted to include other covariates. This means that if precipitation is an important covariate and the modeling framework is used in a simulation rather than a prediction mode, precipitation can be included as a covariate.
We added these points to the discussion section. **See lines 436-441 and 433-434** in the revised manuscript.

4. *Choice of regionally averaged covariates: the location parameter of the Generalised extreme values (GEV) distribution is modelled in a nonstationary way, as a function of time-dependent large-scale climate variables and regional mean variables (accumulated snow water equivalent (SASWE) and April mean temperature (SAAMT)). I have a couple of comments on this choice:*

   a) *Why are the snow and temperature covariates spatially averaged? For such small catchments, wouldn't local covariates (e.g., the snow water equivalent accumulated within each of the 7 catchments) be more skillful to predict (local) streamflow extremes? Would the choice of local covariates improve the performance of the projections for the 7 sites shown in figure A4b and c?*

   b) *The authors state that, for computing the regional average, they considered (and averaged over) all the snow and temperature stations in the UCRB. I would suggest adding a map, showing the location of such stations used for the covariates and/or a table with summary information. Considering that all the 7 sites are located very close to each other in one part of the UCRB (fig.2), are the selected stations of the covariates, representative for the sites where the streamflow is recorded? I also suggest plotting the timeseries of the local*

*covariates (i.e., at each station) together with the regional average and the seasonal streamflow extremes.*

**Response:**

Thank you very much for this comment and for giving us the opportunity to clarify this point. The regional variables are easier to obtain and do not rely on one single station, which may not necessarily stay in operation. For example, suppose in the future, a particular snow gauge is out of operation. In that case, we can keep using the average of the operative snow gauges and still derive the covariate. Figure 2 shows the locations of the 18 snow gauges and three meteorological stations considered for this study. We only considered stations and gauges inside the region of interest and full record for the period of interest (1965-2018; see Figure 2).

[Figure]

**Figure 2**: Streamflow gauges in the Upper Colorado River basin (UCRB) considered in this study. Light blue squares correspond to the snow gauges (18) and purple triangles to the meteorological stations (3) considered in this study.

To demonstrate the suitability of regional averages as covariates for 3-day spring maximum streamflow, we computed the correlation of 3-day maximum flow with regional average SWE,the best (highest correlation) local accumulated SWE until April, and the mean April temperature (MAT) covariate. Figures 3 and 4 in this response to the reviewer display the time series of normalized maximum spring flow with the SASWE (SAMAT) covariate and the best local SWE (MAT) covariate, respectively, for each gauges in the UCRB. The Figures show that regional covariates can capture the inter-annual variability of the spring maximum streamflow without an important reduction of the correlation for most of the station gauges, with only a few exceptions (Figure 3a-b). This was checked for the other lead times as well (not shown here).

In the revised manuscript, the changes are the following:

- We modified **Figure 2** in the revised manuscript.
- We included a summary of this response in **lines 221-231**.
- We included **Figure 5**, which corresponds to Figures 2a and 3a of this response.

[Figure]

**Figure 3:** Times series of normalized 3-day spring maximum streamflow, best SWE covariate, and the spatial average of accumulated snow water equivalent (SASWE) for the seven station gauges. At the bottom of each panel, the correlation of 3-day maximum flow with the best (highest correlation) local SWE and SASWE covariate are displayed.

[Figure]

**Figure 4:** Times series of normalized 3-day spring maximum streamflow, best mean April temperature (MAT) covariate, and the spatial average April mean temperature (SAAMT) for the seven station gauges. At the bottom of each panel, the correlation of 3-day maximum flow with the best (highest correlation) local MAT and SAAMT covariate are displayed.

5. *Line 3: what do the authors mean by 'connected in space'? does it refer to spatial correlation?*

   **Response:**
   Yes, it refers to spatial correlation, and we changed it to "correlated in space." Please, **see line 3** in the revised manuscript.

6. *Line 88-89: what do the authors exactly mean by "the significance of their slope coefficients" posterior PDFs'? Does it mean that they checked whether 0 was included in the 90 or 95% credible bounds of the posterior distribution of the parameters? To the best of my knowledge, tests for significance of trends do not exist in the Bayesian context. Same at line 155.*

   **Response:**
   Yes, we checked whether 0 was included in the 95% credible bounds of the posterior distribution of the parameters, i.e., for each parameter, we checked whether 95% of the sample values were greater (lower) than 0. Please, **see lines 95-96** in the revised manuscript.

7. *Lines 171-173 do not fit well into this paragraph.*

   **Response:**

Thanks for stressing that this paragraph needed some improvement. We changed these lines to "Therefore, using snow information of the basin can provide skillful projections of maximum spring streamflow several months in advance." Please, **see lines 178-179** in the revised manuscript.

8. *Line 193-194: large scale climate indices are used here with their short names and they are defined only later in this section.*

   **Response:**
   Thanks for catching that. We corrected it. Please, **see lines 199-203** in the revised manuscript.

9. *Line 289: the symbol for the Euclidean norm is missing.*

   **Response:**
   Thanks for catching that. We changed these lines to "with Atlantic Multidecadal Oscillation (AMO, McCabe and Dettinger,1999; Enfield et al., 2001; Hidalgo, 2004; Tootle et al., 2005; McCabe et al., 2007; Timilsena et al., 2009; Nowak et al., 2012), Pacific Decadal Oscillation (PDO, Hidalgo, 2004; Tootle et al., 2005; Timilsena et al., 2009), and El Nino Southern Oscillation (ENSO, Redmond and Koch, 1991; Kahya and Dracup, 1994; Rajagopalan et al., 2000; Thomson et al., 2003; Timilsena et al.,2009)". Please, see line 306 in the revised manuscript.

10. *Line 315: what is the benchmark? Is it the stationary (regional) model?*

    **Response:**
    In **lines 313-314**, we included the following "For this study, we considered the climatology (sampling from the observations) as the reference projection and benchmark."

11. *Figure 9: this representation of the results is not really easy to 'read' and it is hard to compare between the 2 panels.*

    **Response:**
    Thanks for the comment. We want to clarify the goal of that figure. Figure 9 is included in the main text because we want to show that adding a Gaussian copula to the model (Figure 9b) can increase the prediction spread (boxes and whiskers of the boxplot). Consequently, the model with a Gaussian copula can capture all the observed values inside the ensemble spread (whiskers), and most of them are inside the 50% credible interval (boxes). This is not possible with the model without a Gaussian copula (red points in figure 9a). To facilitate a comparison between the 2 panels, the two plots have the same limits on the Y-axis. **This figure is the Figure 10 of the revised manuscript**. We included the definition of ensembles in the caption of this figure

12. *Lines 378-404 should not go into the 'discussion' section in my opinion since they are part of the results of the application.*

    **Response:**
    Thanks for the comment. We agree with the referee that this section could also be part of the results section. However, we also felt that since it discusses a future extension of the proposed framework, it was appropriate to show it in the discussion section.

13. *Figure A4: adding vertical lines or white space to separate the 7 sites would be beneficial for the interpretation of the figure*

**Response:**

Thanks for the suggestion. We modified this figure (Figure 5 below). Please, **see Figure A4 in the supplementary material.**

[Figure]

**Figure 5:** The continuous rank probability skill score (CRPS) distribution for different lead times from (a) calibration; (b) leave-1-year-out cross-validation; and (c) leave-1-year-out for extremes (60th percentile) cross-validation. Dark turquoise boxplots denote a 0-month lead time, turquoise boxplots a 1-month lead time, and light turquoise boxplots a 2-months lead time. Higher values of the ESS indicate better model performance. The whiskers show the 95% credible intervals, boxes the interquartile range, and the horizontal lines inside the boxes, the median. Outliers are not displayed. All the lead time models consider a Gaussian copula. As for the ESS, the CRPSS ranges from -∞ to 1, and its values have the same meaning.

**References**

Koster, R. D., Mahanama, S. P. P., Livneh, B., Lettenmaier, D. P., & Reichle, R. H. (2010). Skill in streamflow forecasts derived from large-scale estimates of soil moisture and snow. *Nature Geoscience*, *3*(9), 613–616. https://doi.org/10.1038/ngeo944

Livneh, B., & Badger, A. M. (2020). Drought less predictable under declining future snowpack. *Nature Climate Change*, *10*(5), 452–458. https://doi.org/10.1038/s41558-020-0754-8

Safeeq, M., Shukla, S., Arismendi, I., Grant, G. E., Lewis, S. L., & Nolin, A. (2016). Influence of winter season climate variability on snow–precipitation ratio in the western United States. *International Journal of Climatology*, *36*(9), 3175–3190. https://doi.org/10.1002/JOC.4545

Werner, K., & Yeager, K. (2013). Challenges in forecasting the 2011 runoff season in the colorado basin. *Journal of Hydrometeorology*, *14*(4), 1364–1371. https://doi.org/10.1175/JHM-D-12-055.1

---

## Author Comment (AC2)

**Referee #2**

Thank you for your comments and valuable suggestions. Below are our point-by-point responses to the referee's comments (in italic). We hope they will find them to be comprehensive and satisfactory. In the responses, we refer to specific figures or lines in the main text, to allow the referee to follow the changes implemented in the revised manuscript.

*The topic of the manuscript is certainly within the scope of the journal and the use of a Bayesian Bayesian hierarchical framework for modeling seasonal extremes is groundbreaking; there are no other papers that I am aware of that utilize this approach for this purpose.*

**Response:** Thank you very much for acknowledging the value of our work.

*My few comments are mainly related to terminology, the underlying dataset, and a question about the title and future application of the framework (which, of course, would be beyond the scope of this study).*

1) *L54 and L58: Use of the word "nonstationarity"*
   *In L54, it appears that the term "nonstationarity" is meant in terms of things such as climate change, more akin to what we might think of as long term changes to the system as what non-statisticians think of as only nonstationarity; however, nonstationarity also refers simply to the seasonal signal in the streamflow time series. In L58, the study is asking whether the "representation of nonstationarity through suitable covariates improves season predictions..." Here it appears that you are referring to nonstationarity in the more precise statistical term of nonstationarity. It may be helpful to add a sentence or phrase in the introduction to define nonstationarity in the strict statistical terms so non-statisticians reading the text will not be confused. (I hope I did not confuse things!)*
   **Response:**
   Thanks for the comment. In the paper, we define nonstationarity as temporal variability, i.e., year-to-year variability. However, the framework can also be applied in the climate change context if one has snow projections.
   Based on your suggestion, we modified line 58 to "How does the representation of nonstationarity (inter-annual-variability) through suitable covariates improve seasonal predictions?" **See line 61** in the revised manuscript.

2) *The study uses 7 streamgages to complete the testing of this framework. Looking at Table 1, UCRB7 is an outlier in drainage area, mean streamflow, and mean seasonal streamflow from the other streamgages. It is also located substantially further away from the other streamgages.*
   a. *How does the framework account for streamgages that are outliers. In Figure A4, the model certainly has a different behavior for CHRB7 for the cross-validation in only high flow years. Does this affect the*
   **Response:**
   As shown in Figure 3, the correlation between the stream gauges is similar for all the gauges, and station 7 does not behave as an outlier. We think that the issue that the reviewer noticed in Figure A4 is related to the skill of the covariate for a 1-month lead time rather than the use of the Gaussian copula. This is because all the models in Figure 4 consider a Gaussian copula. The only difference between them is the covariates (station 7's skill for other lead times is similar to those from the other stations). In addition, this decrease in skill

is more related to years with low flows as shown in Figure 4A of the supplementary material which shows higher CRPSS for a year with high flows only.

b. *How robust is your understanding of the spatial dependencies on performance skill when only these 7 streamgages are used? This is a key question you had planned to examine (L59)?*
**Response:**
Figures 8 (7) and 10 (9) in the revised manuscript (manuscript) show that the performance skill can be improved when a copula is added to the model. This means that the model can capture observed values of average maximum specific spring flow inside the ensemble spread of the projected flow (whisker of Figure 10b), which does not happen for the model without a copula (Figure 10a). Therefore, there is an increase in the Energy Skill Score when using the spatial model (Figure 8). All these metrics are multivariate. This model can be implemented for a larger spatial sample without reducing skill but at the cost of increasing computation time.

c. *Could you comment on why a much larger study area or set of streamgages was not used? If a limitation of this framework is that it cannot be applied to large streamgage networks, I wonder what implications this has for its practical applicability.*
**Response:**
In this application, we restricted the study region to the Upper Colorado River Basin (UCRB) since we wanted to ensure that the gauges considered were sufficiently correlated in space, requiring a multivariate modeling framework. There are more stream gauges in the UCRB, but they have missing values for the period 1965-2018. Although this framework can be applied to a more extensive stream gauges network, we do not recommend that since clusters of different streamflow behavior will develop as the region of interest increases. In that case, it is more efficient to fit a model for each cluster than fit a model for the entire region, which will be more computationally expensive. Fitting a model for each cluster allows using different covariates (more skillful) for each cluster too. We mentioned this in the Discussion section. **See lines 443-448** in the revised manuscript.

3) *Title and future applications*
a. *Of particular interest to the Upper Colorado is also the situation of drought prediction. Could this approach be useful for that situation as well? This is beyond the scope of this study but what would be some of the difference in applying the framework to low-flow extremes.*
**Response:**
Thanks for the question. Yes, the framework can be easily applied to predict low-flow extremes using the same marginal distribution (GEV) or another suitable distribution, and identifying suitable covariates. We mentioned this in the discussion. **See lines 432-433** in the revised manuscript.

b. *Along these lines, the title implies that "seasonal streamflow extremes" would mean both tails of the distribution (high and low streamflows); however, flooding is only examined here. Consider changing the title to reflect this.*
**Response:**
Thanks much for the suggestion. We changed the title to "**A space-time Bayesian hierarchical modeling framework for projection of seasonal maximum streamflow**."